# Corticostriatal dynamics encode the refinement of specific behavioral variability during skill learning

Fernando J Santos[1], Rodrigo F Oliveira[1], Xin Jin[2], Rui M Costa[1]*

[1]Champalimaud Neuroscience Programme, Fundação Champalimaud, Lisbon, Portugal; [2]Molecular Neurobiology Laboratory, Salk Institute for Biological Studies, La Jolla, United States

**Abstract** Learning to perform a complex motor task requires the optimization of specific behavioral features to cope with task constraints. We show that when mice learn a novel motor paradigm they differentially refine specific behavioral features. Animals trained to perform progressively faster sequences of lever presses to obtain reinforcement reduced variability in sequence frequency, but increased variability in an orthogonal feature (sequence duration). Trial-to-trial variability of the activity of motor cortex and striatal projection neurons was higher early in training and subsequently decreased with learning, without changes in average firing rate. As training progressed, variability in corticostriatal activity became progressively more correlated with behavioral variability, but specifically with variability in frequency. Corticostriatal plasticity was required for the reduction in frequency variability, but not for variability in sequence duration. These data suggest that during motor learning corticostriatal dynamics encode the refinement of specific behavioral features that change the probability of obtaining outcomes.

## Introduction

Animals have the ability to learn novel motor skills, allowing them to perform complex patterns of movement to improve the outcomes of their actions. Acquiring novel skills usually requires exploration of the behavioral space, which is critical for learning (*Skinner, 1981*; *Sutton and Barto, 1998*; *Grunow and Neuringer, 2002*; *Kao et al., 2005*; *Olveczky et al., 2005*; *Tumer and Brainard, 2007*; *Miller et al., 2010*; *Wu et al., 2014*). It also requires the selection of the appropriate behavioral features that lead to the desired outcomes (*Skinner, 1981*). It has been postulated that the motor system can learn complex movements by optimizing motor variability in task-relevant dimensions, correcting only deviations that interfere with the final output of the action (*Todorov and Jordan, 2002*; *Scott, 2004*; *Valero-Cuevas et al., 2009*; *Diedrichsen et al., 2010*). By optimizing the precision of an action endpoint, for example, humans can perform smooth movements even in the presence of noise (*Harris and Wolpert, 1998*). Selecting task-relevant features and decreasing task-relevant variability might therefore be a critical component of motor learning (*Franklin and Wolpert, 2008*; *Cohen and Sternad, 2009*; *Valero-Cuevas et al., 2009*; *Costa, 2011*; *Shmuelof et al., 2012*).

The reduction of motor variability specifically in relevant domains suggests that the neural activity giving rise to the task-relevant output is selected during learning. However, it is still unclear how the differential refinement of behavioral variability is encoded at the neural level. It has been suggested that cortical and basal ganglia circuits are important for the selection of task-relevant features (*Costa et al., 2004*; *Barnes et al., 2005*; *Kao et al., 2005*; *Olveczky et al., 2005*; *Jin and Costa, 2010*; *Woolley et al., 2014*). Consistently, it has been previously shown that the initial stages of learning have increased behavioral (*Tumer and Brainard, 2007*; *Jin and Costa, 2010*; *Miller et al., 2010*) and

**\*For correspondence:** rui.costa@ neuro.fchampalimaud.org

**eLife digest** Learning a new motor skill typically involves a degree of trial and error. Movements that achieve the desired outcome—from catching a ball to playing scales—are repeated and refined until they can be produced on demand. This process is made more difficult as the activity of individual neurons and muscle fibers can vary at random, and this reduces the ability to reproduce a given movement precisely and reliably.

It has been suggested that the motor system overcomes this problem by identifying those parts of a task that are essential for achieving the end goal, and then focusing resources on reducing the variability in the performance of those parts alone. Santos et al. now provide direct evidence in support of this proposal by recording the activity of neurons in motor regions of the mouse brain as the animals learn a lever pressing task.

By giving mice a food reward each time they pressed the lever four times in a row, Santos et al. trained the animals to press the lever in bouts. The experiment was then slightly modified, so that the mice had to perform the four lever presses more rapidly in order to earn their reward. Consistent with predictions, the average speed of lever pressing initially varied greatly, but this variability decreased as the animals learned the task. By contrast, the total duration of individual bouts of lever pressing—which depends largely on the number of times the mice press the lever—was just as variable after training as before.

A similar pattern emerged for the activity of individual motor neurons in the mouse brain. Whereas their activity initially varied greatly, this variability decreased over training. Moreover, it became increasingly linked to the variability in the speed of lever pressing, but not with the variability in the duration of individual bouts.

The work of Santos et al. has thus shown in real time how the motor system focuses its efforts on reducing variability in those specific parts of a task that are essential for achieving a goal. Without a process called corticostriatal plasticity, by which the motor system adapts, mice could not refine this variability.

neuronal (*Costa et al., 2004*; *Barnes et al., 2005*) variability, but as specific movements are consolidated, neural variability is reduced in these circuits (*Costa et al., 2004*; *Kao et al., 2005*). This suggests that after initial motor and neural exploration, specific patterns are selected and consolidated (*Costa, 2011*). In this study, we investigated if the dynamics of neural activity in cortical and striatal circuits reflect the changes of variability in specific behavioral domains, and if corticostriatal plasticity is critical for the refinement of particular behavior features.

## Results

### Behavior variability is selectively reduced during motor learning

We trained mice to perform a fast lever-pressing task where they were required to press a lever at increasingly higher frequencies, in order to obtain a 20 mg food pellet. After introducing the animals (N = 20) to the behavioral apparatus and 1 day of continuous reinforcement, where each lever-press was reinforced, animals were trained intensively with three daily sessions for 3 days to perform fast lever presses. In the fast press schedules we introduced a covert minimum frequency target, defined by the inverse of three consecutive inter-press intervals (3 IPIs, 4 presses), which increased across sessions from 0 Hz to a maximum of 4.5 Hz (*Figure 1A*; see 'Materials and methods'). The total number of lever presses per minute increased throughout training ($F_{8,152}$ = 41.34, p < 0.0001; *Figure 1—figure supplement 1A*) and animals rapidly started to organize their behavior in self-paced bouts or sequences of lever presses, until there were almost no single presses (*Figure 1C,E* and *Video 1*).

The distribution of the instantaneous lever press frequencies (calculated as the inverse of the each IPI) shows a clear shift from initial sessions, where animals did mostly slow frequency presses (0–0.5 Hz; but already some higher frequency presses of 0.5–4.5 Hz and >4.5 Hz), to latter sessions where the distribution was shifted towards faster pressing speeds (*Figure 1—figure supplement 1C*). A clear multimodal distribution became evident in log scale, with long IPIs (frequencies <0.5 Hz, *Figure 1B*

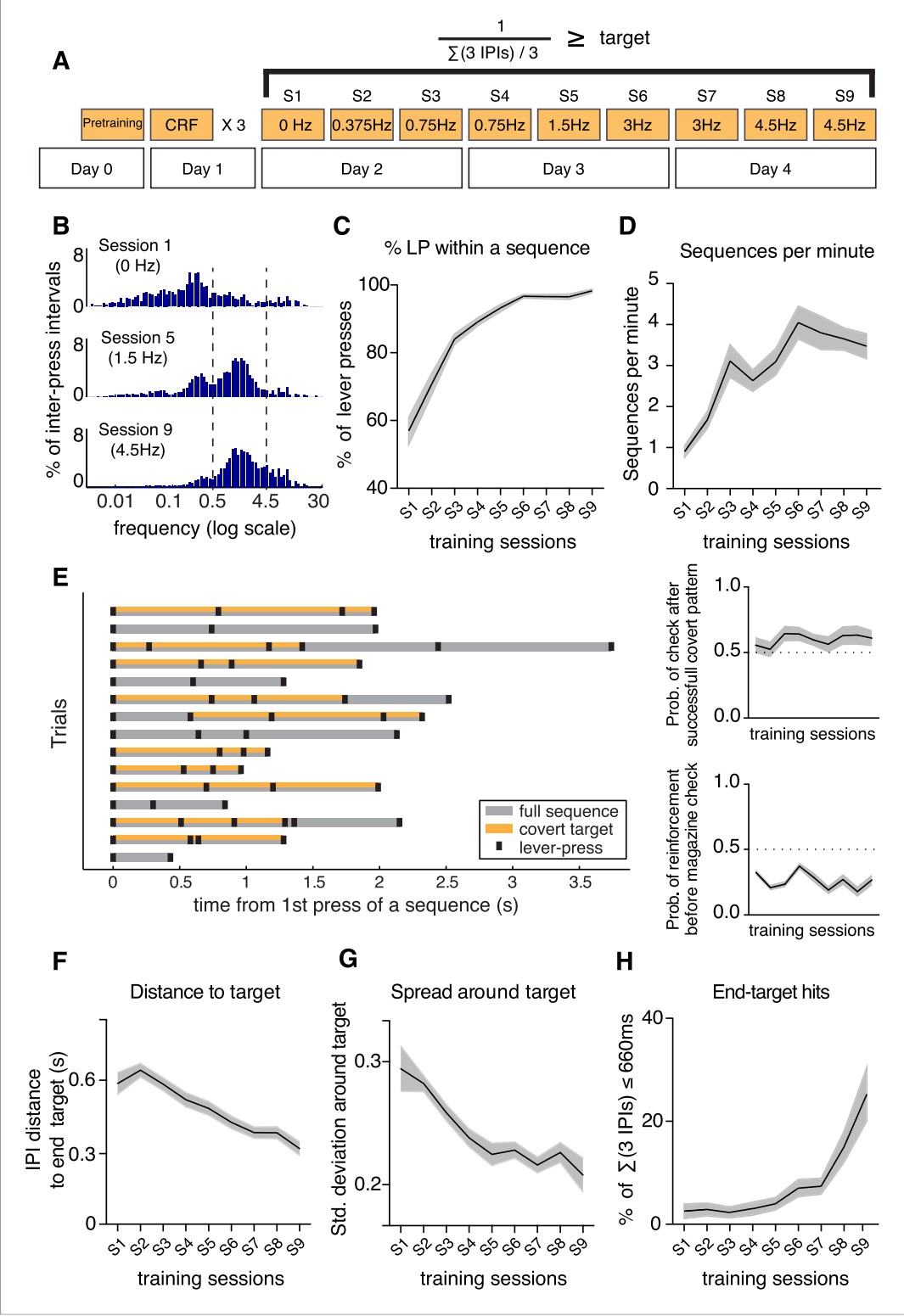

**Figure 1**. Mice learn a fast lever-pressing task, shaping their behavior to gradually approach the minimum frequency target. (**A**) Schematic of the training protocol, starting with magazine habituation and CRF training in the first 2 days, followed by 3 days of the fast press schedules (S1–S9) where we introduce an increasingly higher covert target, defined as the inverse of the sum of three consecutive inter-press intervals (IPIs). (**B**) Joint distribution of the frequency (log scale) for all individual IPIs, in the first, middle and last session of the fast press schedules, for all the

*Figure 1. Continued*

20 animals. Vertical dashed lines correspond to the IPI threshold used for sequence definition (IPI = 2 s, 0.5 Hz) and the final covert target (IPI = 3/660 ms, 4.5 Hz). (**C**) Percentage of lever presses comprised within sequences. (**D**) Number of sequences performed per minute. (**E**) Left: Example of sequences performed by a representative animal, aligned at the time of sequence initiation. Individual lever presses are marked as black ticks, the full sequence duration is shaded in grey and the IPIs that meet the session minimum target are shaded in orange; Top right: Probability of a magazine check immediately after a successful covert target; Bottom right: Probability of a magazine check having occurred after a reinforced lever-press vs a non-reinforced lever-press. (**F**) Distance of all three consecutive IPIs (summed) from the final covert target ($\sum$(3 IPIs) <660 ms, ~4.5 Hz). (**G**) Spread of the distance between all three consecutive IPIs (summed) around the final minimum frequency target. (**H**) Percentage of sequences containing the minimum frequency target of the last session (end-target: 3 IPIs <660 ms, ~4.5 Hz). Shaded areas correspond to mean ± SEM.

The following figure supplement is available for figure 1:

**Figure supplement 1**. Lever-pressing rate increased and shifted towards higher speeds with training, and performance increased or plateaued when task difficulty did not change in consecutive sessions.

and *Figure 1—figure supplement 1D*) representing pauses in pressing or magazine checks. This allowed us to identify the sequences or bouts of pressing a posteriori, based on behavioral performance (either by a pause in pressing higher than 2 s or by the occurrence of checking behavior, i.e., magazine checks between presses; see 'Materials and methods'), independently of the requirements for a specific training session. Importantly, reinforcement delivery did not provide an external cue that could be used by the animals to anticipate a reward, as the probability of performing a magazine check immediately after a successful covert target (instead of performing another press) was not significantly different from 0.5 both on early ($t_{19}$ = 0.9232, p = 0.3675) and late sessions ($t_{19}$ = 1.763, p = 0.0940), and did not change throughout learning ($F_{8,152}$ = 1.753, p = 0.0907, *Figure 1E*, top right). Because a large number of sequences did not contain covert patterns (were not reinforced) we have also calculated the probability of a magazine check having occurred after a reinforced lever-press vs a non-reinforced lever-press, and observed that this was rather low (~0.25) and did not change from early to late sessions (Post hoc comparison: $t_{144}$ = 1.184, p = 0.283, *Figure 1E*, bottom right).

The percentage of lever presses performed within a sequence increased significantly from 56.98 ± 3.98 in the first session of covert target introduction, to 98.26 ± 0.53 in the last training session ($F_{8,152}$ = 60.22, p < 0.0001; *Figure 1C*), and the number of sequences performed per minute increased with training ($F_{8,152}$ = 32.23, p < 0.0001; *Figure 1D*). The percentage of reinforced sequences tended to decrease, since the difficulty of the task increased across sessions, but tended to stabilize or increase when the same target difficulty was repeated in two consecutive sessions ($F_{8,152}$ = 57.31, p < 0.0001; *Figure 1—figure supplement 1B*).

Importantly, with training, the distance of consecutive IPIs (summed in bins of 3 IPIs to mimic the online criteria) to the final target frequency (3 IPIs <660 ms, ~4.5 Hz) decreased consistently ($F_{8,152}$ = 25.76, p < 0.0001; *Figure 1F*), indicating that animals shaped their behavior gradually to approach the end target. Not only did the distance to the end target decrease, but the spread around the target also decreased ($F_{8,152}$ = 9.616, p < 0.001; calculated as the standard deviation around the target frequency, *Figure 1G*). Consistently, animals gradually increased the percentage of press

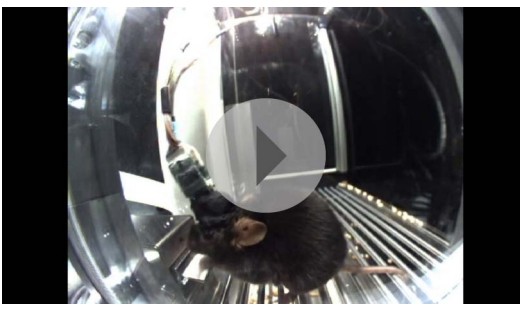

**Video 1.** Animal performing sequences of lever-presses, doing magazine checks and obtaining reinforcement during the last training session. A 20 mg food pellet was delivered in the magazine when the animal performed three consecutive presses within 660 ms (covert target = 4.5 Hz).

bouts that would achieve the minimum target frequency of the last session (end-target: 3 IPIs <660 ms, ~4.5 Hz; $F_{8,152} = 14.15$, p < 0.0001; *Figure 1H*). These data indicate that animals learned to shape their behavior to get closer to the covert target.

The mean frequency of each pressing bout (sequence frequency) decreased slightly ($F_{8,152} = 2.372$, p = 0.0195, *Figure 2A*), while the duration of each pressing bout (sequence duration) increased with training ($F_{8,152} = 22.69$, p < 0.0001, *Figure 2B*). Importantly, the sequence-to-sequence variability of the behavioral parameters (measured both by the variance and by the Fano factor, *Figure 2C–F*) was differentially modulated during training. While the variability of sequence frequency decreased significantly throughout training (variance: $F_{8,152} = 4.450$, p < 0.0001, *Figure 2C*; Fano factor: $F_{8,152} = 5.343$, p < 0.0001, *Figure 2E*), the variability of sequence duration significantly increased (variance: $F_{8,152} = 11.15$, p < 0.0001, *Figure 2D*; Fano factor: $F_{8,152} = 16.86$, p < 0.0001, *Figure 2F*). The sequence-to-sequence variability of these two behavioral features was independent as there was no correlation between the variability in sequence frequency and the variability in sequence duration (variance: $R^2 = 0.0135$; Fano factor: $R^2 = 0.0119$, *Figure 2—figure supplement 1*). This is in contrast with a strong correlation observed between variability in sequence duration and the variability in sequence length—number of presses (variance: $R^2 = 0.8710$; Fano factor: $R^2 = 0.8839$, *Figure 2—figure supplement 1*). The decrease in frequency variability cannot be explained by animals reaching a ceiling in pressing frequency, since the average frequency did not increase with training (it actually decreased slightly). Furthermore, frequency variability started stabilizing after session 4 where the target constrains are still rather loose (3 IPIs in less than 4 s) and this is a frequency that animals can reach in 78.91 ± 5.09% of the sequences at the end of training.

In order to test the specificity of these results, a different group of animals (N = 8) was trained on a control task (*Figure 2H*), where sequences of exactly four consecutive presses were reinforced but where the frequency at which these sequences were performed was not relevant. In contrast with the results observed for the frequency task, in which the sequence-to-sequence variability in frequency decreased ($F_{8,152} = 5.343$, p < 0.0001) and in duration increased ($F_{8,152} = 16.86$, p < 0.0001) (*Figure 2G*), in this control task the variability of sequence frequency did not decrease with training ($F_{8,56} = 1.049$, p = 0.4113), while variability in sequence duration did ($F_{8,56} = 4.589$, p = 0.0002) (*Figure 2H*).

These data indicate that the decrease in variability in sequence frequency was task-specific.

To further investigate this, we analyzed if the variability of these two behavioral dimensions was different in reinforced vs non-reinforced sequences (*Figure 3*). We verified that sequences leading to reinforcement had indeed significantly lower variability in frequency compared to non-reinforced sequences (main effect of reinforcement, $F_{1,38} = 7.608$, p = 0.0089, *Figure 3C* and $F_{1,38} = 28.34$, p < 0.0001, *Figure 3E*), but there were no significant differences in the variability of sequence duration between reinforced and non-reinforced sequences (*Figure 3D,F*). These results suggest that mice selectively reduced variability in the behavioral domains where variability affected the probability of reinforcement (sequence frequency), but not in domains where variability did not change this probability (sequence duration).

## Variability of motor cortex and striatal activity decreases with learning

In order to investigate the dynamics of cortical and striatal circuits during the acquisition and performance of the fast lever pressing task, we continuously recorded extracellular neuronal activity simultaneously in layer 5 of the primary motor cortex (M1), and in the dorsal striatum (DS) of mice during the full duration of training (4 days, N = 7 animals, average of 18 M1 units and 10 DS units simultaneously recorded per animal, per session). Non-stop continuous electrophysiological recordings across 4 days encompassing all the sessions of training allowed us to track the activity of a subset of 'stable' cells throughout the whole period of training (49 M1 units, 21 DS Units). Putative single-units were isolated based on waveform characteristics, inter-spike intervals (ISI) and clustering statistics using principal component analysis (PCA). Units were considered 'stable' if the statistics in PCA space and waveform proprieties did not change significantly across sessions (see 'Materials and methods' and *Figure 4—figure supplement 1C*).

We found a high sequence-to-sequence variability in the activity of individual neurons (measured by the Fano factor of the firing rate) in the first couple of sessions, that then decreased with training (DS: $F_{8,48} = 2.767$, p < 0.05; M1: $F_{8,48} = 2.771$, p < 0.05; *Figure 4A*). These dynamics in neuronal

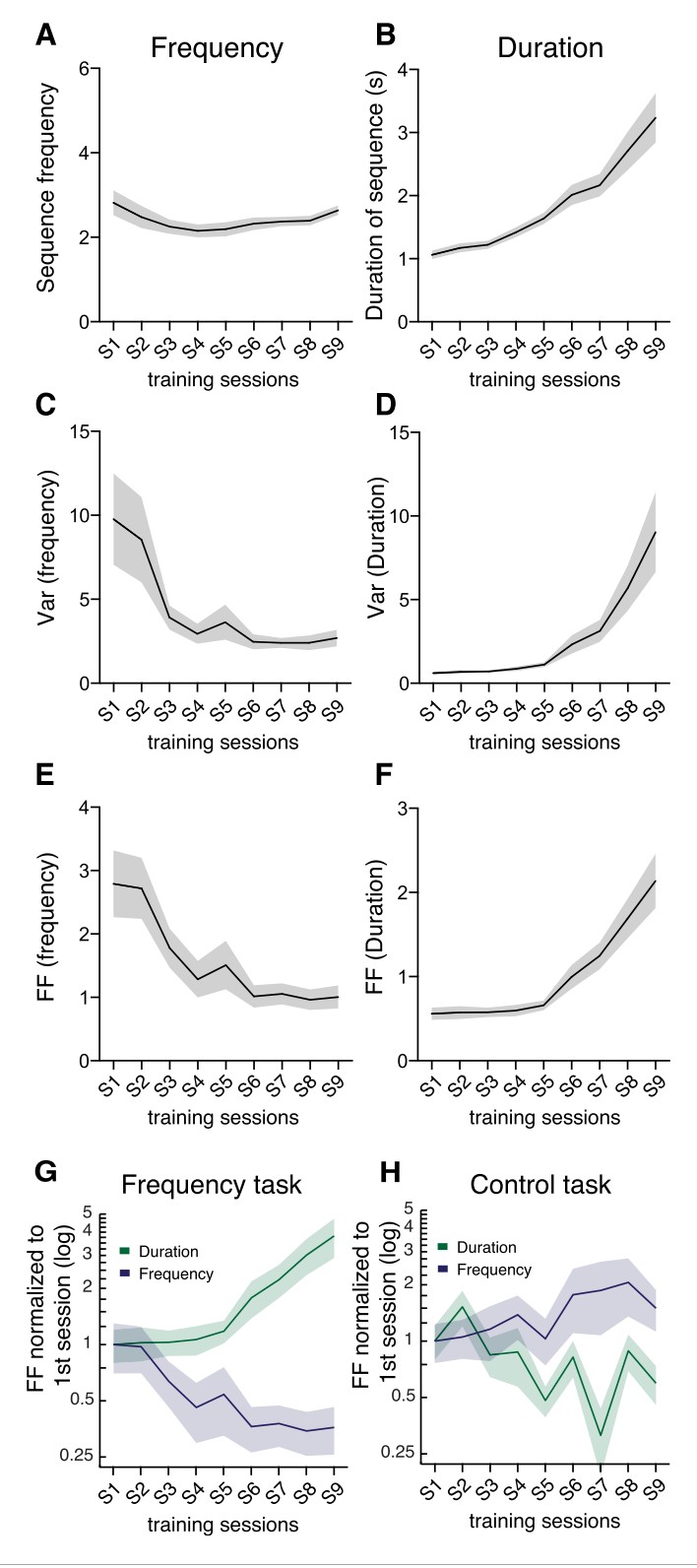

**Figure 2**. Variability of behavioral dimensions evolves independently as animals learn a motor task. (**A**, **B**) Frequency and duration of lever-press sequences (**C–F**) Variability, measured as the variance and Fano factor, for sequence frequency and sequence duration. (**G–H**) Fano factor of both frequency and duration, normalized to the first session, for the frequency and control tasks. Shaded areas correspond to mean ± SEM.

*Figure 2. continued on next page*

*Figure 2. Continued*

The following figure supplement is available for figure 2:

**Figure supplement 1**. Significant correlation between variability of number of presses and duration, but not between variability of frequency and duration.

variability were observed during the performance of lever-press sequences, but not during baseline periods (measured from 5 to 2 s before the initiation of each sequence), when the animals were not actively engaged in lever pressing (DS: $F_{8,48} = 1.117$, $p = 0.3324$; M1: $F_{8,48} = 1.459$, $p = 0.1973$; *Figure 4B*), or during periods flanking the sequence (first press: DS $F_{8,48} = 1.213$, $p = 0.3121$; M1 $F_{8,48} = 0.1374$, $p = 0.9971$; last press: DS $F_{8,48} = 0.5227$, $p = 0.8335$; M1 $F_{8,48} = 0.8677$, $p = 0.5499$; *Figure 4—figure supplement 2*). The decrease in neuronal variability was also observed when using exclusively 'stable' cells for this analysis (DS: $F_{8,160} = 5.223$, $p < 0.0001$; M1 $F_{8,384} = 12.72$, $p < 0.0001$; *Figure 4C*), showing that the differences in variability throughout learning could be observed in individual cells, and did not represent a shift in the population of neurons recorded across days. Importantly, the average firing rate of individual cells did not change significantly, neither across sessions nor across days ($p > 0.05$ for all conditions, *Figure 4E–H*), suggesting that the reduction in variability was not attributable to overall changes in firing rate, but instead to the selection/refinement of a particular firing patterns related to sequence execution.

Further analysis of these dynamics for individual stable cells clearly showed higher variability relative to baseline during the initial sessions (first session DS: $W = 134$, $p = 0.0107$; first session M1: $W = 1119$, $p < 0.0001$), that decreased throughout training until it reached the same levels of baseline at the end of training (last session DS: $W = 73$, $p = 0.2157$; last session M1: $W = 253$, $p = 0.2121$; *Figure 4I*). Again, average firing rates did not show any significant modulation in relation to baseline throughout the whole period of training (DS: $F_{8,160} = 1.031$, $p = 0.4153$; M1: $F_{8,384} = 1.757$, $p = 0.084$; *Figure 4J*).

This decrease in sequence-to-sequence variability of neural activity did not seem to result from the behavior becoming more stereotyped with training, as variability in behavior decreased for frequency but increased for duration (*Figure 2*). To further control that the decrease in neural variability was due to gross changes in behavior we restricted our analyses to sequences matched for frequency ($t_{48} = 1.800$, $p = 0.0781$) and duration ($t_{48} = 1.733$, $p = 0.0895$) between early and late sessions (*Figure 5A,B*). We observed that neuronal variability was still elevated in early sessions and decreased as training progressed (DS: $F_{8,48} = 2.732$, $p = 0.0144$; M1: $F_{8,48} = 2.491$, $p = 0.0239$; *Figure 5C*). Again, these dynamics were not observed during baseline periods (DS: $F_{8,48} = 1.483$, $p = 0.1884$; M1: $F_{8,48} = 1.241$, $p = 0.2965$; *Figure 5D*) and no changes in firing rates were evident in sequence (DS: $F_{8,48} = 0.4684$, $p = 0.8723$; M1: $F_{8,48} = 0.4040$, $p = 0.9128$; *Figure 5E*) or baseline periods (DS: $F_{8,48} = 0.2208$, $p = 0.9855$; M1: $F_{8,48} = 0.3354$, $p = 0.9479$; *Figure 5F*). Single unit analysis also revealed a significant decrease in Fano factor modulation throughout training (DS: $F_{8,160} = 2.688$, $p = 0.0084$; M1:$F_{8,384} = 9.705$, $p < 0.0001$; *Figure 5G*) with no modulation in firing rates (DS: $F_{8,160} = 0.3008$, $p = 0.9648$; M1:$F_{8,384} = 1.406$, $p < 0.1923$; *Figure 5H*).

## Corticostriatal variability becomes correlated with specific behavioral variability

The results above suggest that the decrease in corticostriatal variability is not related to a general decrease in behavioral variability. We therefore investigated if the changes in sequence-to-sequence variability in neural activity were related to the changes in sequence-to-sequence variability of specific behavioral dimensions. We re-calculated the Fano factor of the behavioral features and the neuronal activity using a moving average of a reduced number of trials (5) to provide a higher within session resolution of the variability dynamics and therefore permit the correlation of behavioral and neuronal dynamics across training for each animal (*Figure 6A*, see 'Materials and methods'). Analyses of the relationship between the variability of the recorded units and the variability of each independent behavior feature revealed a significant increase in correlation between neuronal and behavior variability, specific for sequence frequency (*Figure 6C*), but not for duration (*Figure 6D*). These results were observed when using only task-relevant or non-task-relevant neurons (data not shown). They

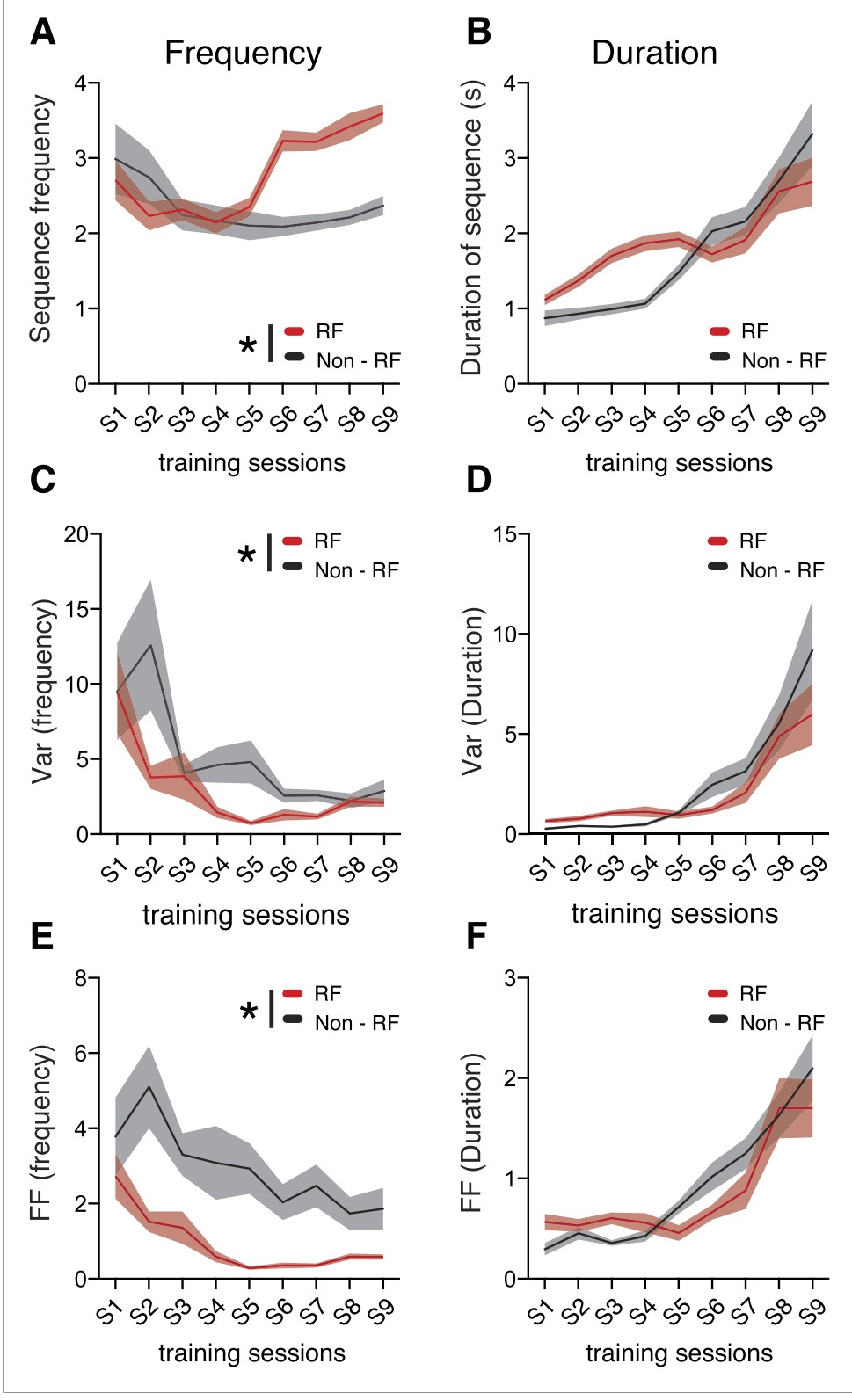

**Figure 3**. Behavior variability is differentially modulated during training. (**A**, **B**) Comparison of frequency and duration between reinforced (RF) and non-reinforced (Non-RF) sequences. (**C**, **D**) Variance and (**E**, **F**) variability, measured as the Fano factor, for reinforced and non-reinforced sequences. Black lines correspond to mean values for non-reinforced sequences. Red lines correspond to mean values for reinforced sequences. Shaded areas correspond to mean ± SEM.

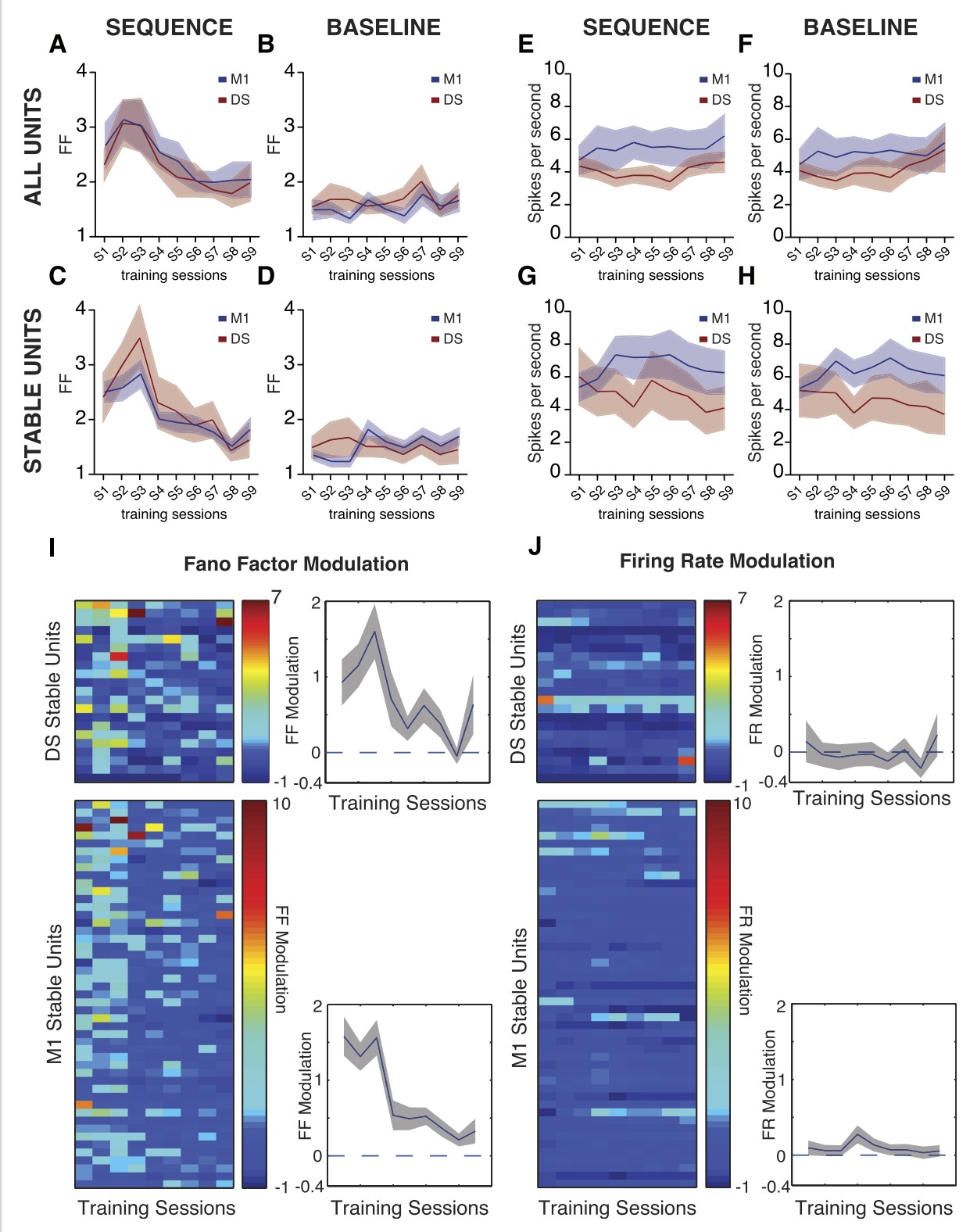

**Figure 4**. Trial-to-trial variability in corticostriatal circuits decreases throughout training. (**A–D**) Neuronal variability (measured as the Fano factor of firing rates) during sequence performance and baseline periods, for all the recorded neuronal units and exclusively for 'stable units', for both M1 (blue traces) and dorsal striatum (DS, red traces). (**E–H**) Firing rates during sequence performance and baseline, for all the recorded units and exclusively for stable units, for M1 (blue traces) and DS (red traces). (**I, J**) Fano factor (FF) and firing rate (FR) modulation relative to baseline values, for individual units recorded

*Figure 4. continued on next page*

*Figure 4. Continued*

across the training sessions (stable units) within DS (top colorplots) and M1 (bottom colorplots). Right panels depict average modulation. Shaded areas correspond to mean ± SEM.

The following figure supplements are available for figure 4:

**Figure supplement 1**. Histological confirmation of electrode tip position and stable units criteria.

**Figure supplement 2**. Neuronal variability around the first and last press of a sequence does not change with training.

were also observed using different number of trials for calculating the moving average of the Fano factor (*Figure 6—figure supplement 2*).

These results show that the decrease in variability in M1 and DS is not just a reflection of a more constrained performance of the movement as training progresses; variability of the movement decreased in a specific dimension but it increased in others were no significant correlation with neuronal variability was evident. Furthermore, no significant correlations were observed between the firing rate of neurons and the variability any of the behavior features (*Figure 6—figure supplement 1*), indicating again that the observed relationship between neuronal and behavior dynamics was not the reflex of a general increase in correlation between neuronal activity and behavior.

The data presented above suggested that as training progressed variability in M1 and striatum became more correlated with variability in a specific domain of behavior that changed the probability of reinforcement. This suggests that neural variability in M1 and striatum could also become more coupled with training. We verified that at the onset of training the sequence-to-sequence variability of neural activity in DS and M1 in each animal was not correlated. However, a strong correlation between the variability in DS and M1 rapidly emerged during training ($p < 0.05$ for all except the first training session, *Figure 6B*), suggesting that as behavioral variability is refined, neural variability in M1 and striatum becomes correlated.

## Corticostriatal plasticity is required for the refinement of behavior variability

The results presented above show that a coupled reduction in corticostriatal variability accompanies the reduction in variability of sequence frequency, but not of sequence duration, suggesting that corticostriatal plasticity is necessary to select the appropriate motor features and hence reduce variability within specific domains. We decided to directly test if the observed reduction in sequence frequency variability is dependent on corticostriatal plasticity by using mutant mice with NMDA receptors deleted specifically at glutamatergic synapses of striatal projection neurons (*RGS9-L^{Cre}::Grin1^{tm1Yql}*; referred to in the figures as striatal projection neuron SPN NR1-KO), which have impaired corticostriatal plasticity (*Dang et al., 2006*), and control littermates. Mutant animals had more difficulty learning the task, so we adapted the training protocol to one session per day for both mutant and littermate controls (and repeated sessions when needed), in order to achieve comparable performance levels (see 'Materials and methods', *Table 1* and *Figure 7A*).

As expected, the distance to target (Controls: $p = 0.0450$, $t_5 = 2.657$, *Figure 7B*) and spread around the target (Controls: $p = 0.0179$, $t_5 = 3.466$, *Figure 7C*) decreased in littermate controls. However, neither of these measures changed with training in mutants (Mutants: $p = 0.3535$, $t_6 = 1.005$; and $p = 0.2817$, $t_6 = 1.183$, respectively; *Figure 7B,C*).

In general, no significant difference was observed for any of the behavior features between the two groups of animals. However, planned comparisons did show that *RGS9-L^{Cre}::Grin1^{tm1Yql}* mutants did not decrease sequence frequency variability during training, in contrast to littermate controls which did (significant main effect of training time: $F_{1,10} = 10.13$, $p = 0.009$; Posthocs: Mutant group: $t_{10} = 1.38$, $p = 0.1964$; Control group: $t_{10} = 3.00$, $p = 0.0134$). Importantly, no differences in the modulation of sequence duration variability were observed between the two groups (no significant main effect for genotype: Duration FF: $F_{1,10} = 0.02$, $p = 0.887$) (*Figure 7D–G*). These statistical results were robust as they were confirmed using bootstrapping statistics (using 100.000 random samples of the data, with replacement) (*Figure 7—figure supplement 1*). These data suggest that corticostriatal

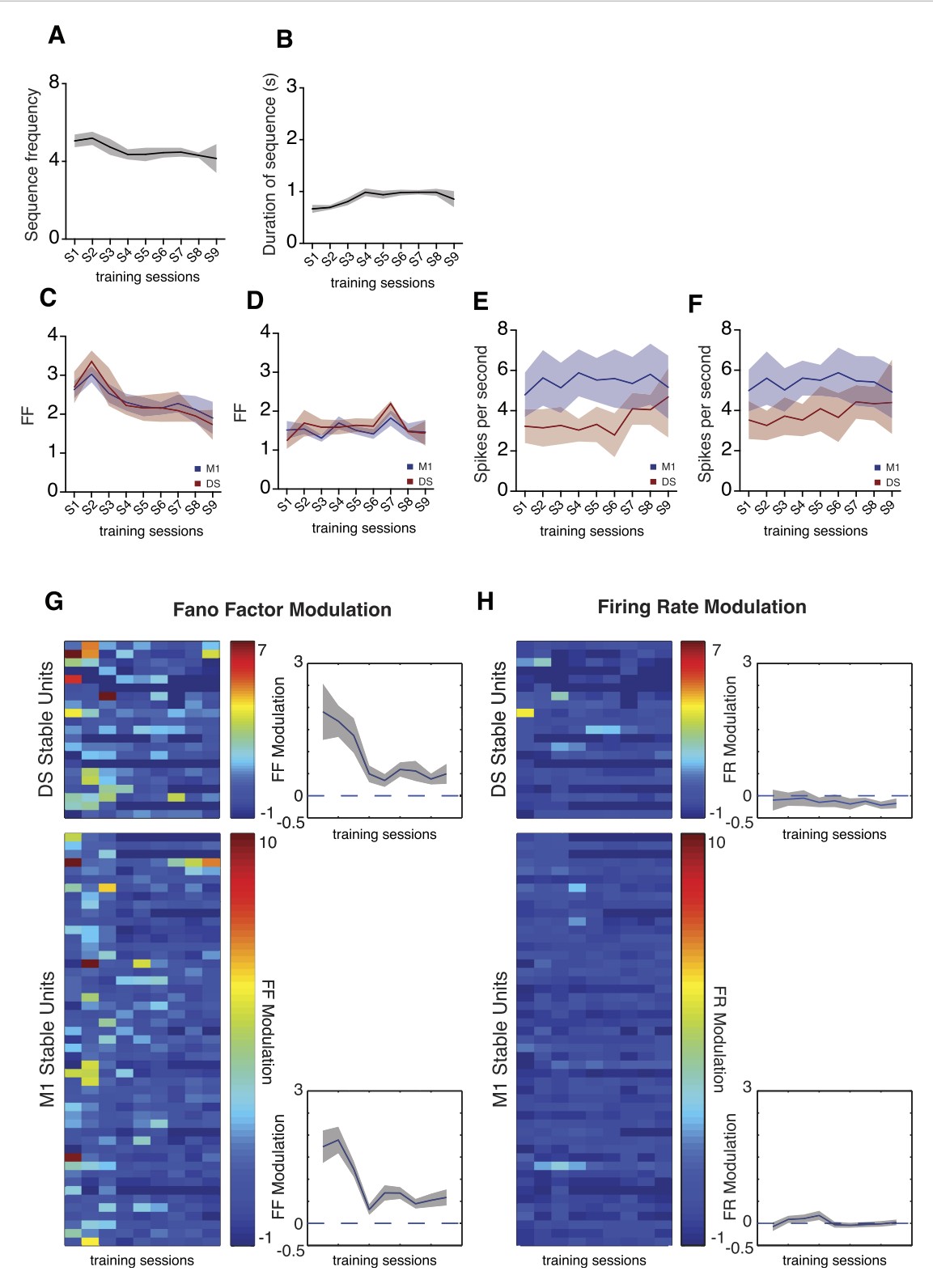

**Figure 5**. Neuronal variability dynamics are still evident when analysis is restricted to sequences with duration and frequency. (**A**, **B**) Frequency and duration of matched sequences. (**C**, **D**) Neuronal variability, measured as the Fano factor of the firing rate, for sequences of matched duration and frequency, for both recorded areas, during sequences and baseline. (**E**, **F**) Firing rates, for sequences of matched duration and frequency, during sequences and baseline. (**G**, **H**) Fano factor (FF) and firing rate (FR) modulation relative to baseline values, for individual units recorded across the training

*Figure 5. continued on next page*

Figure 5. Continued

sessions (stable units) within DS (top colorplots) and M1 (bottom colorplots), for sequences of matched duration and frequency. Right panels depict average modulation. Error bars correspond to mean ± SEM.

plasticity is required for the reduction in variability of specific behavioral features that change the probability of reinforcement.

## Discussion

In this study we show that when mice are trained on a difficult operant paradigm they differentially refine specific behavioral features. When mice were asked to perform progressively faster covert patterns of lever presses to obtain a reinforcer, they reduced variability in sequence frequency, but increased variability in an orthogonal uncorrelated feature (sequence duration). These results are interesting because both features would be classically considered task-relevant—a covert sequence of four presses, which is the minimum to produce a reinforcer in this task, has to have a minimal duration. However, although both features could be considered relevant for the task, only changes in frequency variability were differentially reinforced. Reinforced sequences had lower variability in frequency than non-reinforced sequences, but had equal variability in duration as non-reinforced sequences. Thus, our results indicate that animals reduced frequency variability because that was what was reinforced throughout training. Consistent with this interpretation, in a task where the exact number of presses (correlated with duration) was reinforced but the frequency at which the sequence was performed was not, variability in duration decreased and in frequency increased. This in line with data demonstrating differential modulation of the different components of task space during learning (*Todorov and Jordan, 2002*; *Müller and Sternad, 2004*; *Cohen and Sternad, 2009*).

In previous studies from our group where animals performed operant tasks where the constrains were more relaxed (*Jin and Costa, 2010*), animals decreased variability in all behavioral domains (i.e., they became more stereotypical overall). However, when faced with a more challenging task as in the present study, they decreased variability in the domain that was critical for getting a reinforcer, but increased variability in orthogonal domains (i.e., they were more stereotypical in just a particular domain). It could be that the increase in variability in the orthogonal behavioral domains happens because in difficult tasks animals try to minimize the effort to obtain reinforcers, and hence do not attempt to reduce variability in more than one independent domain. Alternatively, it could also be that mice increased the duration of the sequence (and the correlated number of presses) as a strategy to try to increase the probability of getting a successful covert pattern in that sequence. However, this second possibility is less likely, given that the two behavioral features were not correlated, and that sequences of different durations were equally likely to get reinforced. These data suggest that in more challenging motor tasks it is difficult to reduce variability in all domains, and animals seem to differentially refine the motor patterns that led to reinforcement. Consistently, the number of sequences that comply with the minimum frequency required for the last session (end-target) increased with training and the distance to the end-target decreased with training, indicating that mice implicitly learned to shape their behavior to match the task requirements.

At the neural level, we observed initial high sequence-to-sequence variability of neuronal activity in corticostriatal circuits that decreased with training. Variability in the spike patterns of individual neurons and populations of neurons may be the bases for a process of behavioral exploration (or trial) (*Olveczky et al., 2005*; *Kao et al., 2005*; *Mandelblat-Cerf et al., 2009*), while a decrease in neural variability may reflect a process of selection of specific patterns of neural activity that lead to specific behavioral outputs (*Costa et al., 2004*; *Kao et al., 2005*; *Fee and Goldberg, 2011*). It has been suggested that a decrease in corticostriatal variability as a motor task is learned (*Costa et al., 2004*; *Barnes et al., 2005*) could correspond to the process of selection and consolidation of specific motor patterns (*Costa, 2011*). Here, we show that this decrease in neural variability in corticostriatal circuits correlates specifically with the decrease in variability of a particular behavior domain. These data suggest that the neural patterns in motor cortex and sensorimotor striatum that give rise to the

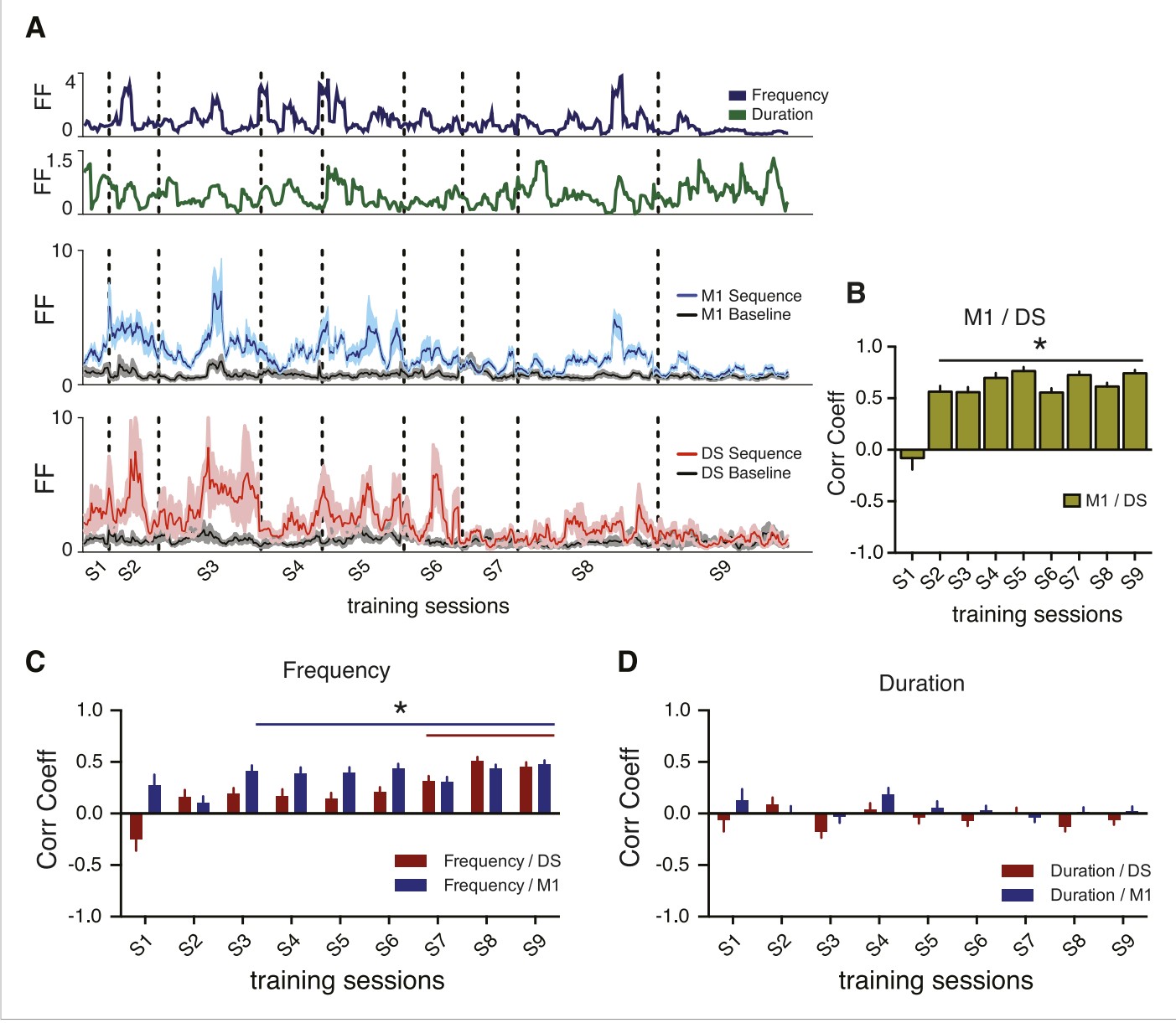

**Figure 6**. Correlations between corticostriatal and behavioral variability emerge for specific behavioral features. (**A**) Example traces from a single animal representing variability, calculated as the Fano factor, using a moving window of five consecutive trials shifted by one for sequence frequency (dark blue trace), sequence duration (green trace), M1 units firing rate during sequences (blue trace) and baseline (grey trace), and DS units firing rate during sequences (red trace) and baseline (grey trace). Vertical dashed lines represent separation of different training sessions. Shaded areas correspond to mean ± SEM. (**B**) Correlation between the variability (FF) in M1 and DS. (**C**, **D**) Correlation between variability traces from neuronal firing rates in M1 (blue bars) or DS (red bars), and variability of sequence frequency or duration. Error bars denote correlation coefficient ±standard error of the correlation. *p < 0.05.

The following figure supplements are available for figure 6:

**Figure supplement 1**. No significant correlation was found between average firing rate and any of the behavior features.

**Figure supplement 2**. Changing the number of trials used for Fano factor calculation did not affect the observed corticostriatal and neuronal/behavioral variability correlations.

behavioral patterns that are reinforced are progressively selected. Provocatively, it also suggests that changes in motor variability that are not specifically reinforced but are part of a strategy or driven by effort reduction may be encoded somewhere else.

**Table 1**. Training protocol and respective number of animals reaching performance criteria for the SPN NR1-KO group and littermate controls

| Training protocol | | Free | 0.375 Hz | 0.375/0.75 Hz (30 reinf) | 0.75 Hz | 1.5 Hz | 1.5/3 Hz (30 reinf) | 3/6 Hz (10 reinf) | 6/7.5 Hz (10 reinf) |
|---|---|---|---|---|---|---|---|---|---|
| # of subjects reaching criteria | NR1–KO | 7 | 7 | 6 | 6 | 5 | 4 | 1 | 1 |
| | Controls | 5 | 5 | 5 | 5 | 5 | 5 | 5 | 2 |

Finally, we also show that corticostriatal plasticity is important for the refinement of specific behavior features. Our data therefore suggests an important role for corticostriatal plasticity in selecting the appropriate implicit neural and behavioral patterns that are reinforced (*Costa, 2011*). However corticostriatal plasticity did not seem to be necessary for the increase in behavioral variability in other domains (*Goldberg and Fee, 2011*). Although in this study we don't investigate the mechanisms underlying the generation of variability, several studies have suggested that the basal ganglia, dopaminergic system, specific cortical circuits, or cerebellar circuits could subserve this function (*Olveczky et al., 2005*; *Costa et al., 2006*; *Leblois et al., 2010*; *Costa, 2011*; *Fee and Goldberg, 2011*; *Shmuelof and Krakauer, 2011*; *Woolley et al., 2014*).

Taken together, our findings suggest that corticostriatal plasticity is important to select the neural patterns that lead to the movement patterns that are reinforced. They highlight that corticostriatal plasticity is not only important for choosing which action to do, but also to shape how to do it to obtain a desired outcome.

## Materials and methods

### Animals

All experiments were carried in accordance to the ethics committee guidelines of the Champalimaud Foundation and Instituto Gulbenkian de Ciência, and with approval of the Portuguese DGAV (Ref. 0421). Experiments were carried out using 20 male, 3 to 5 month old C57BL6/J mice. From these, 13 animals were used exclusively for behavioral training while the remaining seven underwent microelectrode array implantation for neuronal data recordings. Animals were maintained on a light–dark cycle of 12 hr:12 hr starting at 7 AM. All experiments were done during the light cycle. Mice were housed in groups of four animals prior to surgery and individually after the electrodes were implanted. 3 to 6 months old *RGS9-L^Cre^::Grin1^tm1Yql^* homozygous mice (N = 7) and *Cre* negative littermate controls (N = 5) were used for the mutant mouse behavioral experiments.

### Surgery and in vivo extracellular recordings

Seven C57Bl6/J mice were implanted bilaterally with two micro-electrode arrays (2 × 8), 35–50 μm tungsten electrodes with micro-polished tips. One array targeted the primary motor cortex (M1, layer 5) while the second was targeting the (DS, sensorimotor area that receives projections from the same area in M1). Craniotomies and electrode array positioning were done according to coordinates from the Mouse Brain Atlas (*Paxinos and Franklin, 2008*). M1 array was placed 1 mm rostral and 1.6 mm lateral from bregma, and lowered ~1 mm from the surface of the brain. DS array was placed 0.5 mm rostral and 2.1 mm lateral from bregma, and lowered ~2.3 mm from the surface of the brain. Electrodes were manually lowered at slow rates while constantly monitoring neural activity in all the channels in order to control for proper electrode function and correct positioning. Final verification of electrode position was done after all the experiments were finished, by perfusing animals with PFA and histological confirmation of Nissl stained 70 μm brain slices (*Figure 4—figure supplement 1A,B*). After surgery animals were allowed to recover for at least 2 weeks before starting any other experimental procedure. Single and multi unit activity was recorded using Blackrock Microsystems Neural Signal Processor, allowing for online sorting of identified units. Further offline sorting of selected units was done using Plexon Offline Sorter v3 (Plexon Inc, Dallas, TX, United States), based on waveform characteristics, ISI and PCA clustering. Units stability was assessed from waveforms and PCA cluster proprieties. For PCA cluster

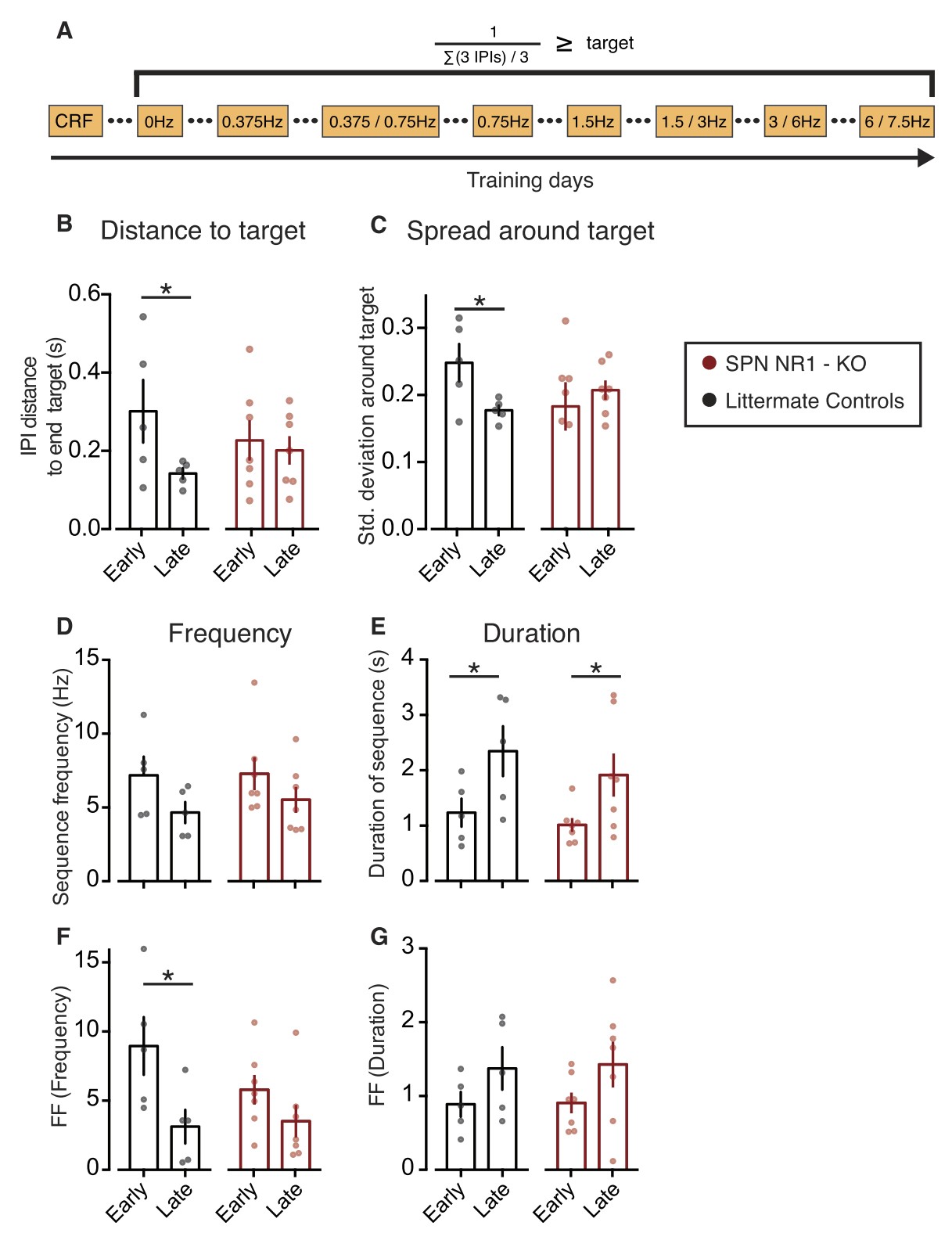

**Figure 7**. Corticostriatal plasticity is necessary for the specific refinement of behavioral variability. (**A**) Schematic of the adapted training sessions for mutant animals and littermate controls. Animals would remain in the same training session until reaching a stable performance. (**B**) Distance of the sum of all three consecutive IPIs from the final covert target (∑(3 IPIs) <660 ms, ~4.5 Hz) in SPN NR1 mutants and littermate controls (**C**) Spread of the distance between three consecutive IPIs around the final covert target. (**D**–**G**) Behavior parameters and variability, measured as the Fano factor, during early and

*Figure 7. continued on next page*

*Figure 7. Continued*

late training sessions in SPN NR1 mutants and littermate controls groups. Bars correspond to mean, with data from individual animals plotted on the background (red: SPN NR1-KO; black: littermate controls).

The following figure supplement is available for figure 7:

**Figure supplement 1**. Bootstrapping statistics in the SPN NR1-KO data support the observations from the post hoc planned comparisons.

comparison data from all the training sessions was pooled together to calculate common eigen vectors. Data from individual sessions was then projected into this common PC space, allowing us to determine cluster centroids and dispersion for each session. Clusters were considered stable whenever the centroid in a given session was comprised within the interval of the centroid of the previous session $\pm 1.96$ * standard deviation of the cluster, in the first two principal components (*Figure 4—figure supplement 1C* for a graphical representation of this criteria).

## Behavioural training

Animals were trained using operant chambers (MedAssociates Inc, St. Albans, VT, United States) placed inside sound attenuating boxes. A retractable lever was extruded in the beginning of each session, simultaneous to the onset of a light. Animals were required to perform a sequence of presses at a minimum frequency in order to obtain a 20 mg food pellet (Bio-Serv, Flemington, NJ, United States). 24 hr before the first training session animals were placed under a food restriction schedule. Body weight was constantly monitored in order to be kept above 85% of the initial weight. In order to facilitate learning, animals were initially exposed to one session of magazine training were food pellets would be available on a random time schedule, and to three sessions of continuous reinforcement schedule (CRF) 1 day before training, where single lever presses would be reinforced. On the following training sessions animals were reinforced if they performed a sequence of consecutive presses at a minimum frequency (covert target), defined by the inverse of three consecutive inter-press intervals (IPIs), which increased with training. On the first session there was no minimum frequency target, meaning that any consecutive 3 IPIs would lead to reinforcement. In consecutive sessions the minimum frequency that would lead to reinforcement was increased or maintained in the following order: 0.375 Hz, 0.75 Hz, 0.75 Hz, 1.5 Hz, 3 Hz, 3 Hz, 4.5 Hz and 4.5 Hz. This constant increase in the minimum frequency of the covert target forced the animals to systematically adapt to the task requirements and perform faster sequences of presses from session to session. The training protocol for mutant animals and littermate controls was adapted due to difficulties learning the task, to one daily session and using automatic progressive schedules once a minimum number of reinforcements (30 or 10) was achieved. (*Table 1* for performance summary.)

## Sequences of lever presses

Sequences of presses were differentiated based on IPI and occurrence of a magazine head entry. An IPI >2 s (determined based on the distribution of IPIs) or a head-entry were used to define the bouts or sequences of presses. The 2 s cutoff was determined from the joint distribution of the instantaneous IPIs (and the corresponding log distribution) from all the animals, by determining the valley between the two main peaks of IPIs (*Figure 1—figure supplement 1C,D*). Frequency of each sequence was defined as the inverse of the average IPI of each sequence. Duration of each sequence was defined as the time between the first and the last press event. Length of each sequence was defined as the number of press events in each sequence. For the matched sequences analysis, sequences with a duration of 0.2–2 s and a frequency higher than 2 Hz were selected.

## Task-related neurons

Neural activity was averaged in 20-ms bins, shifted by 1 ms, and averaged across trials to construct the peri-event histogram (PETH). Data from the PETH from 5000 to 2000 ms before lever press were considered as baseline activity. A positive modulation in firing rate was defined if at least 20 consecutive bins had firing rate larger than a threshold of 99% above baseline activity, and a negative modulation of firing rate was defined if at least 20 consecutive bins had a firing rate smaller than a

threshold of 95% below baseline activity (*Belova et al., 2007*). Paired t-tests between baseline firing rate and sequence firing rate were used to classify individual neurons as sequence-related.

## Analysis and statistics

The programs to run the tasks presented in this study can be found at http://tinyurl.com/or7ug72. Analyses were done in Matlab (MathWorks, Natick, MA, United States) or GraphPad Prism (GraphPad Software Inc, La Jolla, CA, United States). Normality was verified for all tests using the D'Agostino-Pearson omnibus normality test, or the Kolmogorov–Smirnov test when sample size was too small. Repeated measures ANOVA were used to evaluate changes in behavior and neuronal features. Probability of a magazine check after lever-press was evaluated using one-way ANOVA and post hoc comparisons using Fisher's LSD test, but one subject was excluded from these analysis due to a lack of recorded timestamps for magazine head-entries. Paired t-tests were used to evaluate differences in percentage of lever-presses. Increases in FF modulation were assessed by the Wilcoxon Rank Signed test. Repeated measures two-way ANOVA was used to verify the general effect of the *RGS9-NR1* mutants experiment. Bootstrapping statistics were used on the data from the *RGS9-NR1* mutants and littermate controls to validate the results from the post hoc tests. Histograms were built from 100000 randomized samples with replacement. Sample sizes were calculated based on $\alpha = 0.05$ and power of 0.7. Trial to trial variability of neuronal and behavior data was assessed using Fano factor. We calculate the Fano factor of individual units by dividing the variance of firing rates across all the trials of a session by the mean over those trials. Fano factor and firing rate modulations for individual stable cells were calculated as the ratio between the difference of values for sequence and baseline and the values during baseline (Fano factor: $[FF_{sequence} - FF_{baseline}]/FF_{baseline}$; firing rate: $[FR_{sequence} - FR_{baseline}]/FR_{baseline}$). Fano factor of the behavioral features was calculated by dividing the variance in the individual features by the mean of the feature for all the trials. To establish correlations between the variability of the neuronal data and the variability of the behavior, Fano factors were calculated using three, five or seven consecutive trials, allowing us to increase the resolution of the variability measures. Correlations between neuronal and behavior data were evaluated using Pearson's linear correlations. To avoid correlations bias due to sample size, statistical significance of all the correlations was assessed using the significance criteria for the session with smaller size. Within animal correlations averaged using Fisher's z transformation (*Silver and Dunlap, 1987*) returned similar results to grouped correlations for all the tested conditions (data not shown).

## Acknowledgements

We thank V Paixão and A Gomez-Marin for valuable comments on the manuscript and A Vaz for animal colony management. This research was supported by the INDP Graduate Programme and a FCT fellowship to FJS, and European Research Council Consolidator Grant, HHMI International Early Career Scientist Grant, and ERA-Net NEURON grants to RMC.

## Additional information

### Competing interests

RMC: Reviewing editor, *eLife*. The other authors declare that no competing interests exist.

### Funding

| Funder | Grant reference | Author |
| --- | --- | --- |
| Howard Hughes Medical Institute (HHMI) | International Early Career Scientist Grant IEC 55007415 | Rui M Costa |
| European Research Council (ERC) | Consolidator Grants, ERC CoG 617142 | Rui M Costa |
| ERA NET | NEURON | Rui M Costa |

The funders had no role in study design, data collection and interpretation, or the decision to submit the work for publication.

## Author contributions
FJS, Conception and design, Acquisition of data, Analysis and interpretation of data, Drafting or revising the article; RFO, XJ, Acquisition of data, Drafting or revising the article; RMC, Conception and design, Analysis and interpretation of data, Drafting or revising the article

## Author ORCIDs
Rui M Costa, http://orcid.org/0000-0003-1707-1051

## Ethics
Animal experimentation: All experimental procedures were carried in accordance to the ethics committee guidelines of the Champalimaud Foundation and Instituto Gulbenkian de Ciência, and with approval of the Portuguese DGAV (ref 0421).

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
