## [Decision Letter]

[Editors’ note: a previous version of this study was rejected after peer review, but the authors submitted for reconsideration. The first decision letter after peer review is shown below.]

Thank you for choosing to send your work entitled “Corticostriatal dynamics encode the refinement of outcome-relevant variability during skill learning” for consideration at *eLife*. Your full submission has been evaluated by Timothy Behrens (Senior Editor) a member of the Board of Reviewing Editors and three peer reviewers and the decision was reached after extensive discussions between the reviewers. Based on our discussions and the individual reviews below, we have reached the decision that we will reject the paper as is.

The reviewers agree that the study addresses an important issue in neuroscience and is of potential interest for a broad audience. There are, however, strong concerns whether the main claims of the paper are supported by the data as presented. An elaborated and unbiased analysis is needed to show that the behavior and electrophysiological results indeed support the main claims in the study. Because of *eLife's* policy direct invitation for revision should not require elaborated work we are forced to reject the paper. However, we would be willing to look at a new manuscript that addresses the main concerns that were raised against the study. It is essential that, in particular the point about outcome-relevant specificity, is addressed appropriately.

The main elements of concerns are outlined here and further elaborated in the specific comments from the reviewers.

1) The analysis should provide clear evidence that the reduction in frequency variability is in fact real and that the animal is refining its behavior. The authors need to demonstrate that the behavior is indeed refined and that the variance will decrease not just because the duration of presses decreases or that the number of presses in each sequence is very small in the first sessions. It also should be clear that the effect is not just the out of sequence presses that decrease rather than the in sequence that increase. For analysis of the neuron data is should be clear that on what segment the firing rate is measured. If the firing rate is measured on the entire duration of the pressing sequence then it is liable to the same criticism as pointed out above. Namely, measuring the firing rate in longer time windows will have smaller variance.

2) The main part of the story rests on the claim that it is only variability in the outcome relevant aspect of the behavior that decreases with learning. This is in contrast to variability in sequence length or duration, allegedly outcome-irrelevant aspects. Yet the data suggest that longer sequences improve task-performance and that the mice increase them with learning. Thus length/duration seems to be outcome relevant, despite the experimental protocol not explicitly rewarding on these features.

This point needs to be substantiated with further analysis to show an outcome-relevant specificity. It will be important to discuss why behaviors under FR3/0.66s protocol in this study and FR4/0.5s protocol in earlier study give rise to very different behavioral patterns. The authors should include a re-analysis of the FR4/FR8 behavioral results in this paper to show that, under slightly different operant requirements, mice can selectively reduce the variability of press length instead of press frequency. They should directly test the simple possibility that M1/DS activity linearly encode press frequency (for example average press frequency of a sequence; or max frequency of a sequence; or instantaneous frequency associated with each press) using correlation analysis. If such is the case, the authors should quantify the overlap between sequence-related neurons and press-related neurons, and see if the two populations show more overlap over training blocks. Alternatively, the absence of significant correlation would suggest that M1/DS activity is coding for properties related to press frequency in non-linear ways, and FF correlation is a novel approach to reveal this hidden relationship. As additional controls to establish the specificity of the observed FF correlation, the authors should (1) clearly indicate whether this analysis involve all neurons, or only sequence-related neurons, (2) indicate what the time window used to calculate average firing rate within a press, (3) provide a correlation analysis done on a per-neuron basis, (4) indicate if lever-press related neurons show the same correlation, as well as what happen to other task-unrelated M1/DS neurons.

3) The number of animals in Figure 7 seems very small, there are no error bars and the effects seem to be governed, in some cases by 1-2 animals. The authors should demonstrate that the main result is not due to these animals only.

Reviewer #1:

In this paper, Santos et al. investigated whether motor variability in the outcome-relevant dimension specifically reduces during learning, and whether such reduction is mediated by neuronal activity in corticostriatal circuits. By requiring mice to increase peak press frequency in order to obtain reward, the authors found that variability in press frequency is selectively decreased. Such a reduction in behavioral variability is correlated with a concomitant reduction in M1/DS neuronal activity variability over learning blocks, and abolished in mice with deficient corticostriatal plasticity. These results are potentially very interesting in that they provide elegant experimental support for a widely-held prediction in motor control literature, and also provide a new conceptual framework to analyze behavioral and neuronal dynamics during learning when the mapping between neuronal activity and behavior continues to evolve. I have a number of concerns that I wish the authors address using existing data.

1) A big part of the story rests on the behavioral finding that the variability of press frequency decreased while the variability for press length and duration increased. I was initially concerned that there may be a trivial explanation of this result, or that any animal undergoing press-related operant training may show similar behavior. However, after comparing the current results with those in Xin and Costa (2010) and Xin, Tecuapetla, and Costa (2014), I think it is likely the case that, in earlier studies, mice learn to optimize the number of press under extended FR4 or FR8 training protocol, and thus specifically reduced variability for the number of presses.

I think it is very important for the authors to emphasize this comparison and discuss why behaviors under FR3/0.66s protocol in this study and FR4/0.5s protocol in earlier study give rise to very different behavior patterns. To that end, the authors should perhaps include a re-analysis of the FR4/FR8 behavioral results in this paper to show that, under slightly different operant requirements, mice can selectively reduce the variability of press length instead of press frequency.

2) The moving-window FF correlation between the behavioral features and neural activity (Figure 6) is fascinating. This analysis shows that M1/DS activity FF (but not baseline activity FF) was correlated specifically with FF of average press frequency, but not length or duration. The use of FF to investigate the evolution of neural coding during learning is very clever because, without knowing exactly what behavioral parameters M1/DS activity encode and how the encoding may evolve during learning, the FF correlation strongly supports that M1/DS activity must encode properties related to the average press frequency.

To elaborate on this observation, the authors should directly test the simple possibility that M1/DS activity linearly encode press frequency (for example average press frequency of a sequence; or max frequency of a sequence; or instantaneous frequency associated with each press) using correlation analysis. The presence of significant correlation indicates that M1/DS neurons are encoding for press frequency. If such is the case, the authors should quantify the overlap between sequence-related neurons and press-related neurons, and see if the two populations show more overlap over training blocks. Alternatively, the absence of significant correlation would suggest that M1/DS activity is coding for properties related to press frequency in non-linear ways, and FF correlation is a novel approach to reveal this hidden relationship.

As additional controls to establish the specificity of the observed FF correlation, the authors should: (1) clearly indicate whether this analysis involve all neurons, or only sequence-related neurons; (2) specify what is the time window used to calculate average firing rate within a press (i.e. first press to the last press); can the authors include another window around the onset of the press sequence (i.e. [-1s to first press] or [-0.5s to 0.5s] of first press)?; (3) specify what was the correlation analysis done on a per-neuron basis and then averaged across all neurons (Figure 6) and neuron pairs (Figure B)?; and (4) clarify whether lever-press related neurons show the same correlation. What about other task-unrelated M1/DS neurons? One would hope not.

Reviewer #2:

The paper claims that mice reduce performance variability during performance improvement on a lever-pressing task, but only in the outcome-relevant dimension. They then go on to show that trial-by-trial variability in the average firing rate correlates with performance variability in the outcome-relevant dimension, and more so late in learning. They then use a knockout mouse to probe whether variability reduction requires cortical input to the striatum.

Figure 1 plots the sequence rate, and 2A the sequence frequency. I assume that these are the same, but why are the graphs so different? Then the authors switch to talking about variability in pressing frequency, which is something different. But if we assume that what they call sequence frequency in 2A is in fact pressing frequency (I have no idea if that's the case, but it's a fair assumption given the rest of the paper), then I become confused, because Figure 1—figure supplement 1 clearly shows that instantaneous pressing frequency is increased with learning, yet 2A suggest otherwise.

The authors suggest that sequence frequency is task-relevant but that the length of the sequence is not. Yet their own data (Figure 3) shows that the longer the sequence the more reward is being delivered, hence from the mouse's point of view sequence length seems to be a relevant dimension. If one wanted to make the claim that the mouse decreases the variability in the task relevant dimension over other task-irrelevant ones, then one should design a task that has two explicit and comparable dimensions and make reward contingent first on one and then the other in separate experiments. For example, one could make the first interval in 3-tap sequence subject to some reward criteria, but the second interval not, and then switch it up in the next experiment. If variability decreases for the relevant interval but not the irrelevant one no matter which the reward was contingent on, then that would, to me at least, be a far more compelling result. As it stands they are comparing dimensions that both seem relevant for reward, one which has an increase in variability with learning, another which has a decrease.

Further down, why do the authors compare all neurons when they look at neuronal variability during the task? It seems to me that this analysis should be done only on task-related neurons.

There are also other confounds, like reward probability decreasing with learning, something that on its own is known to affect motor variability.

Reviewer #3:

The manuscript suggests that the refinement of behavior in success related dimensions is correlated with refinement in corticostriatal spiking patterns. This is an important point to be made and the type of experiment the authors did is suitable. The idea itself is not completely novel and several previous studies have suggested this and shown reduction in variance as learning progresses; nevertheless, this is a nice demonstration of the concept and therefore can be important to the field. I do have some concerns with the analyses that dampen my enthusiasm and raise questions about the interpretation of the results. The authors need to make a cleaner and unbiased analysis to show that the behavior and electrophysiological results indeed support the main claim.

Summary of substantive concerns:

1) Are the animals really refining their behavior? The major changes along the training process seem to be:

1.1) The percentage of lever presses within sequences increase (Figure 1). Alternatively the ‘out of sequence’ presses decrease (Figure 1—figure supplement 1, all points left of -0.3 dashed line);

1.2) The mean number of presses in a sequence increases (Figure 2);

1.3) The sequence frequency increases (Figure 1).

The mean frequency in the sequences does not change much (Figure 2

A). This, in fact, may raise concerns about the change in Figure 2. If the IPI in the sequences is drawn (i.i.d.) from a distribution with some fixed (μ) mean, it follows that the distribution of the press rate in some duration has a related mean (1/μ) but also, due to the central limit theorem, that the variance will decrease with the duration. Therefore, the authors need to find a controlled way to demonstrate that the behavior is indeed refined and not that the animals simply make longer pressing sequences and hit their targets by chance.

I think some simple controlled analyses can be done to address this (e.g. sampling similar periods of time etc.), but it has to be shown.

2) The mean number of presses in each sequence is very small in the first sessions (<3, Figure 2) and the Fano Factor that approaches 1 (Figure 2) suggests just that. Also, the distributions between the dashed lines in Figure 1—figure supplement 1 do not seem to change much in sessions 3-8. It is therefore important to show that the main result is not due to this effect alone (the low number of presses in sessions 1-2).

3) The case for independence of behavioral dimensions in not clear enough. Figure 2—figure supplement 1 is claiming so but it is not entirely clear to me what is the main conclusion. It needs to be better explained.

4) It is not clear on what segment the firing rate is measured? This is crucial. If the firing rate is measured on the entire duration of the pressing sequence then it is liable to the same criticism as point 1 above. Namely, measuring the firing rate in longer time windows will have smaller variance. Having a mutual cause may explain the result in Figure 6.

5) The number of animals in Figure 7 seems very small, there are no error bars and the effects seem to be governed, in some cases by 1-2 animals. The authors should demonstrate that the main result is not due to these animals only.

6) In general, the definition of refinement is a bit over-stated here. Reduction is variability is one aspect, yet I could think of several other approaches to define ‘refinement’ that could be interesting as well and produce a richer manuscript with more interesting conclusions.

[Editors’ note: what now follows is the decision letter after the authors submitted for further consideration.]

Thank you for submitting your work entitled “Corticostriatal dynamics encode the refinement of outcome-relevant variability during skill learning” for peer review at *eLife*. Your submission has been favorably evaluated by Timothy Behrens (Senior Editor), a Reviewing Editor, and three reviewers.

The reviewers have discussed the reviews with one another and the Reviewing Editor has drafted this decision to help you prepare a revised submission.

In this manuscript, Santos et al. investigated whether variability of specific task parameters are reduces during learning, and whether such reduction is mediated by neuronal activity in corticostriatal circuits. By requiring mice to learn a task that involve increased press frequency in order to obtain reward, the authors find that the variability of press frequency is a decreased meter while variability of press duration is not. They conclude that the animals learn to reduce the variability of the frequency as outcome-relevant parameter while outcome irrelevant parameters, like duration, is not changed. They find that the reduction in frequency variability is correlated with a concomitant reduction in M1/DS neuronal activity variability over learning blocks, and abolished in mice with deficient corticostriatal plasticity.

The manuscript is a resubmission. There were several concerns whether the main claims of the manuscript as submitted before were supported by the data as presented. In particular further evidence to support that outcome specificity was restricted to press frequency was requested. The authors have provided new analysis and a set of new experiments to meet these concerns. However, while the reviewers agree that the study has improved after revision the issue of outcome-specificity is still not resolved. Given the paper's focus, this is a major issue that must be addressed. After extensive discussion among reviewers, and editors there is an agreement that the task as presented does not isolate frequency as the only causal dimension in the task. The fact that reward is dispensed based on frequency does not preclude other relevant behavioral parameters, even if those are orthogonal to, or uncorrelated with, sequence frequency. In fact, both sequence frequency and duration are modulated by learning. Success on the task is therefore likely to also depend on duration since it is coupled to sequence length. Therefore, the task does not isolate frequency as the only task-relevant parameter, and the dichotomy between task-relevant (frequency) and task-irrelevant (duration) parameters does not hold. Hence it cannot be claimed that learning is only tuning frequency as the outcome-relevant parameter. Duration is relevant by default in reward accumulation tasks. This may not be reflected in the tuning of neuronal firing patterns while frequency may in this task. This distinction may be interesting but is not spelled out in the manuscript. Careful wording is needed to clarify this and to discuss how the two outcome-relevant parameters that are being compared, sequence frequency and duration, differ. This is important because the difference in the neural correlates of these task aspects, and how they change with learning, is not due to one being task-relevant and the other not, but rather to these being qualitatively different aspects of the task. As the text is now this is not the message the reader will be left with. Thus the dimension along which the two task-relevant parameters differ should be discussed, and an attempt to generalize the results beyond this task should be made. The authors must revise their statements carefully to reflect this and explicitly explain the confounding factor introduced by press duration. This new message should also be reflected in the title and in a more nuanced description in the Abstract, and Introduction of the outcome relevant concept as well as in an expanded discussion of these issues.

A number of other issues were also raised by the reviewers, as outlined below in the detailed report. In particular whether animals know that they are rewarded (see review *#*2). An analysis based on behavioral data should clarify whether the mice indeed learn to anticipate reward. If so this is also a task-relevant parameter that is learned by the animals.

Reviewer #1:

In this paper, Santos et al. investigated whether motor variability in the outcome-relevant dimension specifically reduces during learning, and whether such reduction is mediated by neuronal activity in corticostriatal circuits. By requiring mice to increase peak press frequency in order to obtain reward, the authors found that variability in press frequency is selectively decreased. In contrast, in a control task that required 4 consecutive presses to obtain reward, variability in press duration but not press frequency decreased. The reduction in outcome-relevant behavioral variability is correlated with a concomitant reduction in M1/DS neuronal activity variability over learning blocks, and abolished in mice with deficient corticostriatal plasticity.

These results are potentially very interesting in that they provide elegant experimental support for a widely-held prediction in motor control literature, and also provide a new conceptual framework to analyze behavioral and neuronal dynamics during learning when the mapping between neuronal activity and behavior continues to evolve. I have a number of concerns that the authors should address.

1) The resubmitted manuscript has significantly improved with the addition of the new control experiment. The pattern of behavioral refinement in the control task is the opposite of the main task, which provides strong support that the behavioral refinement reported in this manuscript is specific to the outcome-relevant dimension. The authors should provide further information of this control task to support this key point, by providing un-normalized results of the control task as in Figure 2, as well as comparison between rewarded and non-rewarded trials as in Figure 3.

2) In prior studies, Xin and Costa indicated that mice could not hear reward delivery and waited until the end of press sequence to check for reward. Is that still the case in this study? From Figure 1, it seems that mice commonly check for reward right after reaching the press frequency criteria. The authors should provide an analysis to show how many presses do mice continue to press after reaching the criteria frequency in each press. A reduction of this quantity over training sessions will support that mice gained some knowledge of the reward criteria.

This analysis is also important to address lingering concerns about whether press length/duration is an outcome-relevant dimension. If both length and frequency are relevant, mice should generate long sequences with occasional fast presses. This scenario should predict that fast presses can take place anywhere in the sequence. On the other hand, if press frequency is the only outcome-relevant dimension, the fast presses should occur mostly toward the end of the sequence.

Reviewer #2:

My main concern with the initial submission was that the authors pitted two aspects of the task against each other: frequency and duration. While both are clearly correlated with success and both change with learning, the authors called one (frequency) task relevant and the other (duration) not relevant. I think this is misleading and incorrect. This was pointed out in the previous referee report, and the authors revised manuscript does not seem to address this issue.

The authors say that the two aspects aren't correlated with each other, and that this somehow means that if one dimension is task-relevant (frequency) the other (duration) can't be. At least that is how I infer their logic, but this does not make sense. Independent and uncorrelated aspects of a behavior can of course both be task-relevant.

The mice have to press the lever 4 times in a specified time span. Initially this window is long and they get reward easily, so no need for long durations. Then the task becomes harder. If they go to the reward magazine with the same duration, they now get less reward, but if they increase their duration, the chance of getting 4 presses in the allotted time increases, so they learn to increase the duration while also decreasing the frequency. Both are relevant for the task.

The authors also introduce a ‘control’ that shows that the variance in frequency goes down in the original task because it is task-relevant. I never doubted this. Frequency is task relevant, but so is duration. The control is irrelevant for the point I was making.

There are other issues I raised that the authors did not respond to, e.g. do animals know when reward is available, etc. (their analysis on this point is not addressing the point). The authors should show that the probability of mice going to the reward port is not influenced by the reward dropping into the magazine. The data they refer to in this regard (Figure 3) actually seems to suggest that mice do learn this. Early, the rewarded trials are longer than unrewarded, later they are shorter (3B). This is consistent with the mice ‘learning’ when a reward is available either by picking up on a cue or having an internal sense.

Reviewer #3:

The authors resubmitted the manuscript after doing considerable additional analyses and a control experiment. I find the paper to be fairly convincing now, as the authors addressed the concerns in a serious manner. The results now point to reduction in variability along with improvement in a motor task. This constitutes a nice finding. I hope the authors can convince us further by supplying:

1) Figure 2 (var of freq) computed on equal durations from all the sessions. Perhaps I missed it, but I did not see a clear demonstration that the reduction in freq is not related to the increase in duration. There is nice indirect evidence, and even a control experiment, and they are fairly convincing – but a direct demonstration would be appreciated.

2) Is there any relationship/correlation across individuals between the behavioral improvement and the physiological finding? This would strengthen the study.

[Editors' note: further revisions were requested prior to acceptance, as described below.]

Thank you for resubmitting your work entitled “Corticostriatal dynamics encode the refinement of behavioral variability during skill learning” for further consideration at *eLife*. Your revised article has been evaluated by Timothy Behrens (Senior Editor), a Reviewing Editor, and three reviewers.

In the previous decision letters clear guidance was given for how presentation and discussion should be improved so the paper conveys a clear message supported by the data. A substantial effort was placed in the discussion among reviewers to reach a consensus about the necessary changes. There is a feeling amongst all three reviewers that this has not yet happened and that you have chosen to be selective in your response thereby missing the opportunity to improve the manuscript and meet the raised criticism. We would however like to give you a chance to revise the manuscript so that it meets the raised criticism.

Three main issues still need to be considered:

You were asked to acknowledge that both features you look at (frequency and duration) are outcome-relevant. You have done this only in part. The Discussion mostly uses the old and misleading narrative. This should be remedied.

You were asked to parse the distinction between frequency and duration, and speculate as to why the neural firing patterns associated with these features evolve differently during learning. It was clearly stated that duration and frequency, though both outcome-relevant, are fundamentally different aspects of the task. This needs to be discussed. Frequency has to do with the action that is being reinforced while duration is a strategic decision. The fact that neuronal correlates associated with these fundamentally different processes evolve differently is perhaps not surprising. However, there is no discussion of these differences and what they mean in a form that allows one to generalize to other tasks and situations.

Finally, you were asked to demonstrate whether mice learn to anticipate the reward. You say in the text that you looked at the “probability of a magazine check after a reinforced lever-press” and found that this did not change with learning. In the figure and associated legend you say that you looked at the “Probability of a reinforcement preceding a magazine check” and show that this doesn't change. It is not clear whether these are the same metrics and how they are calculated and whether they allow one to infer anything about the animal's ability to anticipate reward. We advise that you simply show that the probability of checking the magazine does not depend on whether the preceding lever press was reinforced (i.e. the last one in a covert sequence) or not, and show that this is stable over the course of learning.

---

## [Author Response]

[Editors’ note: the author responses to the first round of peer review follow.]

*The reviewers agree that the study addresses an important issue in neuroscience and is of potential interest for a broad audience. There are, however, strong concerns whether the main claims of the paper are supported by the data as presented. An elaborated and unbiased analysis is needed to show that the behavior and electrophysiological results indeed support the main claims in the study. Because of* eLife*'s policy direct invitation for revision should not require elaborated work we are forced to reject the paper. However, we would be willing to look at a new manuscript that addresses the main concerns that were raised against the study. It is essential that, in particular the point about outcome-relevant specificity, is addressed appropriately.*

The main elements of concerns are outlined here and further elaborated in the specific comments from the reviewers.

1) The analysis should provide clear evidence that the reduction in frequency variability is in fact real and that the animal is refining its behavior. The authors need to demonstrate that the behavior is indeed refined and that the variance will decrease not just because the duration of presses decreases or that the number of presses in each sequence is very small in the first sessions.

First of all we would like to apologize if some of the points raised were not clear in the previous version. We had indeed checked that variability in sequence frequency was orthogonal to variability in duration and length of the sequences. However, variability in duration and length (number of presses) was very correlated which may have caused some confusion. In order to simplify and better clarify the behavioral refinement observed we have modified the current version of the manuscript to compare only two main features of behavior that are orthogonal and independently modulated (frequency and duration).

As depicted in the plots in Figure 2—figure supplement 1 (each dot represents one session of each animal, with darker dots corresponding to later sessions), there is no significant correlation between the variability of sequence frequency and variability in sequence duration (measured both by the variance and Fano factor). By contrast, variability of sequence duration is highly correlated and dependent on the variability of number of presses of each session. This observation supports the idea that the behavioral dynamics and modulations observed are independent for the two dimensions described throughout the manuscript (Frequency and Duration). The same lack of correlation happens between variability in the number of presses and the frequency of pressing, as we had indicated in the previous version of the manuscript.

Furthermore, we have performed analyses in sequences with matched behavioral features (see Figure 5 of the current version of the manuscript). Finally, we ran a new control task to demonstrate that the decrease in variability in frequency is because frequency is outcome relevant and not only because the number of press in the sequences changes with learning.

It also should be clear that the effect is not just the out of sequence presses that decrease rather than the in sequence that increase.

We are sorry for the confusion but *all* analyses presented in the previous version of the manuscript exclude out of sequence presses (with the exception of the analyses in Figure 1 and Figure 1—figure supplement 1, that were used to determine the thresholds for sequence criteria). So to be clear, all behavioral and neuronal measurements are done in lever-presses that are part of a sequence of two or more presses. Also, to further clarify, as depicted in Figure 1, the percentage of lever-presses that is within a sequence very rapidly reach values close to 100%.

For analysis of the neuron data is should be clear that on what segment the firing rate is measured. If the firing rate is measured on the entire duration of the pressing sequence then it is liable to the same criticism as pointed out above. Namely, measuring the firing rate in longer time windows will have smaller variance.

Firing rate was measured on the entire duration of the sequence for all the analyses presented in the manuscript. It is pointed out that this could lead to changes in variance that could be dependent on the duration of the sequences. To control for this, we also measured firing rate in 200ms bins shifted by 1ms. The results from measuring the firing rate and Fano factor in fixed width bins (Figure 8) were no different from the measurement on the entire duration of the sequence, hence pointing to the fact that the changes in variance observed were not dependent on sequence duration.

Author response image 1.**DOI:**
http://dx.doi.org/10.7554/eLife.09423.021

Furthermore, there was no correlation between firing rate during sequence

performance and sequence duration, or variability in firing rate and variability in sequence duration.

2) The main part of the story rests on the claim that it is only variability in the outcome relevant aspect of the behavior that decreases with learning. This is in contrast to variability in sequence length or duration, allegedly outcome-irrelevant aspects. Yet the data suggest that longer sequences improve task-performance and that the mice increase them with learning. Thus length/duration seems to be outcome relevant, despite the experimental protocol not explicitly rewarding on these features.

This point needs to be substantiated with further analysis to show an outcome-relevant specificity. It will be important to discuss why behaviors under FR3/0.66s protocol in this study and FR4/0.5s protocol in earlier study give rise to very different behavioral patterns. The authors should include a re-analysis of the FR4/FR8 behavioral results in this paper to show that, under slightly different operant requirements, mice can selectively reduce the variability of press length instead of press frequency.

We would like to first clarify that variability in number of presses or sequence duration was not different between reinforced and non-reinforced sequences (see Figure 3). We took the advice of the reviewers seriously and to further test the hypothesis that features are refined based on the relevance to the outcome of the task we ran a second experiment (Control task) were mice were reinforced if they performed a sequence of exactly 4 presses (3 IPIs), irrespectively of the frequency. In contrast with the results observed for the frequency task, in which the Fano factor for frequency decreased and for duration increased, in this control task variability of sequence frequency (which is not a relevant feature) increases with training, while variability of sequence duration decreases with training (see Figure 2).

This supports the view that, in similar tasks, slight changes in the requirements to achieve reinforcement can lead to opposite modulation of the variability of behavioral features dependent on their relevance for obtaining an outcome.

(Note: in the previous task mentioned, FR4/0.5s variability – measured as cv

– also decreased for frequency/IPIs, as shown in that study. Therefore the new task ran here is a more appropriate control for the questions raised by the reviewers).

They should directly test the simple possibility that M1/DS activity linearly encode press frequency (for example average press frequency of a sequence; or max frequency of a sequence; or instantaneous frequency associated with each press) using correlation analysis. If such is the case, the authors should quantify the overlap between sequence-related neurons and press-related neurons, and see if the two populations show more overlap over training blocks. Alternatively, the absence of significant correlation would suggest that M1/DS activity is coding for properties related to press frequency in non-linear ways, and FF correlation is a novel approach to reveal this hidden relationship.

We had done this but it was probably buried in the complexity of the manuscript. We tested the possibility that neuronal activity can encode sequence frequency (or duration) by studying the correlation between firing rate and the behavior features. As shown in Figure 6—figure supplement 1, there was no significant correlation between the firing rate and any of the measured features of behavior. As suggested by the reviewer, this excludes the simple explanation that firing rate in these structures is largely encoding simple kinematic parameters, and points to the hypothesis that cortico-striatal activity might be coding for outcome-relevant features in non-linear ways.

As additional controls to establish the specificity of the observed FF correlation, the authors should (1) clearly indicate whether this analysis involve all neurons, or only sequence-related neurons.

All analyses presented in the manuscript include all the neurons recorded in each session.

(2) Indicate what the time window used to calculate average firing rate within a press,

Firing rate was calculated using the time window of each sequence, but as clarified above, we have controlled this by using a 200ms window shifted by 1ms, which lead to comparable results.

(3) Provide a correlation analysis done on a per-neuron basis.

Besides the correlations done with the average Fano factor for the neuronal data (Figure 6), we have also calculated the correlations between behavioral variability and neuronal variability on a per-neuron basis (see Figure 9). Despite the overall comparable results these results should be interpreted with care since skewed distributions of correlation coefficients can lead to biases in statistics (Corey, David M, Dunlap, William P, Burke, Michael J, Journal of General Psychology, Vol 125(3), Jul 1998, 245-262). Therefore we prefer to present the results in the manuscript per animal (and mention just in the text).

Author response image 2.**DOI:**
http://dx.doi.org/10.7554/eLife.09423.022

(4) Indicate if lever-press related neurons show the same correlation, as well as what happen to other task-unrelated M1/DS neurons.

This is a good point that we should have clarified. Comparable correlations between neuronal variability and behavior variability were observed when dividing the data between task-related and non-task related neurons (Figure 10). Due to the absence of a specific effect between conditions and groups of neurons, all the analyses used in the manuscript include all neurons recorded, regardless of the firing rate modulations during sequence performance. But we now mention in the manuscript that the results are comparable between task-related and non-task related neurons.

Author response image 3.**DOI:**
http://dx.doi.org/10.7554/eLife.09423.023

*3) The number of animals in*
Figure 7
*seems very small, there are no error bars and the effects seem to be governed, in some cases by 1-2 animals. The authors should demonstrate that the main result is not due to these animals only.*

We believe that it is dangerous to assume by eye that statistical effects are dominated by a few animals. If distributions are not significantly different it still may happen that some data points look different every time we sample the population, and vice-versa. What we try to assess is if the distributions that those samples came from are different or not. Therefore, we did a bootstrapping analysis for all the comparisons in Figure 7 (see Figure 7—figure supplement 1), performing 100.000 random samples of our data (with replacement). These bootstrap analyses confirmed the effects already described by the post hoc tests. Furthermore, we have now introduced error bars in Figure 7 (error bars depict standard error of the mean, albeit in the case of paired comparisons they are not indicative of the variability in the comparison).

[Editors' note: the author responses to the re-review follow.]

[…] The manuscript is a resubmission. There were several concerns whether the main claims of the manuscript as submitted before were supported by the data as presented. In particular further evidence to support that outcome specificity was restricted to press frequency was requested. The authors have provided new analysis and a set of new experiments to meet these concerns. However, while the reviewers agree that the study has improved after revision the issue of outcome-specificity is still not resolved. Given the paper's focus, this is a major issue that must be addressed. After extensive discussion among reviewers, and editors there is an agreement that the task as presented does not isolate frequency as the only causal dimension in the task. The fact that reward is dispensed based on frequency does not preclude other relevant behavioral parameters, even if those are orthogonal to, or uncorrelated with, sequence frequency. In fact, both sequence frequency and duration are modulated by learning. Success on the task is therefore likely to also depend on duration since it is coupled to sequence length. Therefore, the task does not isolate frequency as the only task-relevant parameter, and the dichotomy between task-relevant (frequency) and task-irrelevant (duration) parameters does not hold. Hence it cannot be claimed that learning is only tuning frequency as the outcome-relevant parameter. Duration is relevant by default in reward accumulation tasks. This may not be reflected in the tuning of neuronal firing patterns while frequency may in this task. This distinction may be interesting but is not spelled out in the manuscript. Careful wording is needed to clarify this and to discuss how the two outcome-relevant parameters that are being compared, sequence frequency and duration, differ. This is important because the difference in the neural correlates of these task aspects, and how they change with learning, is not due to one being task-relevant and the other not, but rather to these being qualitatively different aspects of the task. As the text is now this is not the message the reader will be left with. Thus the dimension along which the two task-relevant parameters differ should be discussed, and an attempt to generalize the results beyond this task should be made. The authors must revise their statements carefully to reflect this and explicitly explain the confounding factor introduced by press duration. This new message should also be reflected in the title and in a more nuanced description in the Abstract, and Introduction of the outcome relevant concept as well as in an expanded discussion of these issues.

We sincerely apologize for the misunderstandings that our use of outcome- relevant variability introduced. We believe that these are misunderstandings and do not affect the main message of the paper. So we carefully revised the manuscript as suggested, and abandoned the outcome-relevant/outcome- irrelevant nomenclature to refer to the different behavior dimensions. We now say that animals trained in a task to perform progressively faster sequences of lever presses reduced variability of sequence frequency but increased variability in an orthogonal domain (sequence duration as training progressed, variability in corticostriatal activity decreased and became progressively more correlated with behavioral variability, but only for sequence frequency. Corticostriatal plasticity was required for the reduction in frequency variability, but not for variability in sequence duration. We believe that these statements are not controversial.

We understand how some of the reviewers may find that duration is task- relevant, as any sequence of presses requires a minimum duration. What we meant to say is that, in our task, trial-to trial variability in frequency changes the probability of animals getting reinforcement, while trial-to-trial variability in sequence duration (which is orthogonal to frequency) does not affect the probability of getting a reinforcement. For the range of variability in duration observed in the task presented, there is no difference in variability between reinforced and non-reinforced trials; meaning that reinforced trials have the same variance/variability in duration as non-reinforced trials (and actually no difference in duration per se) (Figure 3). In contrast, reinforced trials do have much lower variability in frequency than non-reinforced trials (Figure 3). It is therefore plausible that animals reduced the variability in the domain that changed the probability of reinforcement. Therefore, in order to clarify this message and our claims, we have carefully revised and altered the title, Abstract and main text of the manuscript.

A number of other issues were also raised by the reviewers, as outlined below in the detailed report. In particular whether animals know that they are rewarded (see review #2). An analysis based on behavioral data should clarify whether the mice indeed learn to anticipate reward. If so this is also a task-relevant parameter that is learned by the animals.

We apologize for not having introduced this before but we thought we just had to address the comments in the summary. In order to clarify this point, we present additional analyses on the behavioral data (See Figure 11 and Figure 1). We have calculated both the probability of a reinforcement preceding a magazine check for individual sessions (see Figure 11, left plot), and the percentage of magazine checks that follow reinforced lever presses vs. non-reinforced lever presses for all the training data (see Figure 11, right plot). The probability of a magazine check after a reinforced lever-press was rather low (∼0.25) and did not change from early to late sessions (Post hoc comparison: t_144_=1.184, p=0.283, Figure 1, right) and the percentage of magazine checks following reinforced presses was significantly lower than for non-reinforced presses (t_19_=12.10, p<0.0001, Figure 1, right).

Both these analysis provide support to the idea that reinforcement delivery does not provide an external cue that could be used by the animals to anticipate a reward.

Author response image 4.**DOI:**
http://dx.doi.org/10.7554/eLife.09423.024

Furthermore, as requested by the reviewers we further show no correlation between sequence frequency and sequence duration per se (Figure 2—figure supplement 1), and a strong correlation between sequence frequency and sequence length (showing that duration and length are not orthogonal).

[Editors' note: further revisions were requested prior to acceptance, as described below.]

Three main issues still need to be considered:

You were asked to acknowledge that both features you look at (frequency and duration) are outcome-relevant. You have done this only in part. The Discussion mostly uses the old and misleading narrative. This should be remedied.

You were asked to parse the distinction between frequency and duration, and speculate as to why the neural firing patterns associated with these features evolve differently during learning. It was clearly stated that duration and frequency, though both outcome-relevant, are fundamentally different aspects of the task. This needs to be discussed. Frequency has to do with the action that is being reinforced while duration is a strategic decision. The fact that neuronal correlates associated with these fundamentally different processes evolve differently is perhaps not surprising. However, there is no discussion of these differences and what they mean in a form that allows one to generalize to other tasks and situations.

We have modified substantially the Discussion to incorporate these two points very clearly. We also changed sentences in the main text that could allude to the old narrative. We feel though that we must first clarify that we initially used the term outcome-relevant feature as a different term than what is usually meant as task-relevant; we used it to mean what the reviewers call the feature that was reinforced. Given the confusion that this has generated we have eliminated references to outcome-relevant in the text and just refer a couple of times to the more established concept of task-relevant feature when discussing the literature.

So now we mostly just refer to the features that were specifically reinforced and others that could be part of a strategy/effort minimization, and discussed it in the terms suggested by the reviewers.

As you can see in the new Discussion, we acknowledge that both features can be considered task-relevant, and further discuss why variability may decrease in one and increase in the other given their relation to reinforcement. We also discuss why the neural activity would correlate with one and not the other, and generalize to other tasks.

Finally, you were asked to demonstrate whether mice learn to anticipate the reward. You say in the text that you looked at the “probability of a magazine check after a reinforced lever-press” and found that this did not change with learning. In the figure and associated legend you say that you looked at the “Probability of a reinforcement preceding a magazine check” and show that this doesn't change. It is not clear whether these are the same metrics and how they are calculated and whether they allow one to infer anything about the animal's ability to anticipate reward. We advise that you simply show that the probability of checking the magazine does not depend on whether the preceding lever press was reinforced (i.e. the last one in a covert sequence) or not, and show that this is stable over the course of learning.

Again, we apologize for the misunderstanding. Because in many sequences there is no reinforcement (or successful covert pattern), we feel that we have to present both probabilities. So we present both the probability of checking the magazine after a successful covert sequence; and the probability of a magazine check being preceded by a successful covert sequence (because many sequences do not contain successful presses the two probabilities are not similar). This should clarify the issue that animals are not checking the magazine because they hear the reinforce-delivery device. Please see the following passage in the main text: “Importantly, reinforcement delivery did not provide an external cue that could be used by the animals to anticipate a reward […] and did not change from early to late sessions (Post hoc comparison: t_144_=1.184, p=0.283, Figure 1, bottom right)”.